# Magnitude and associated factors of anti-retroviral therapy adherence among children attending HIV care and treatment clinics in Dar es Salaam, Tanzania

Fatima M. Mussa [1]*, Higgins P. Massawe[2], Hussein Bhalloo[3], Sibtain Moledina[4], Evelyne Assenga[1]

1 Department of Paediatrics and Child Health, Muhimbili University of Health Sciences, Dar es Salaam, Tanzania, 2 Department of Paediatrics and Child Health, Muhimbili National Hospital, Dar es Salaam, Tanzania, 3 Sanitas Hospital, Dar es Salaam, Tanzania, 4 Department of Internal Medicine, Muhimbili University of Health and Allied Sciences, Dar es Salaam, Tanzania

* f_mussa@icloud.com

## Abstract

### Introduction

The HIV pandemic continues to contribute significantly towards childhood mortality and morbidity. The up-scaling of the Anti-retroviral therapy (ART) access has seen more children surviving and sanctions great effort be made on ensuring adherence. Adherence is a dynamic process that changes over time and is determined by variable factors. This necessitates the urgency to conduct studies to determine the potential factors affecting adherence in our setting and therefore achieve the 90-90-90 goal of sustainable viral suppression.

### Objectives

To assess the magnitude and associated factors of ART adherence among children (1–14 years) attending HIV care and treatment clinics during the months of July to November 2018 in Dar es Salaam.

### Methods

A cross-sectional clinic-based study, conducted in three selected HIV care and treatment clinics in urban Dar es Salaam; Muhimbili National Hospital (MNH), Temeke Regional Referral Hospital (TRRH), Infectious Disease Centre- DarDar Paediatric Program (IDC-DPP) HIV clinics during the months of July to November 2018. HIV-infected children aged 1–14 years who had been on treatment for at least six months were consecutively enrolled until the sample size was achieved. A structured questionnaire was used for data collection. Four-day self-report, one-month self-recall report and missed clinic appointments were used to assess adherence. Frequencies and percentages were used to describe categorical data. The odds ratio was used to analyse the possible factors affecting ART adherence Logistic regression models were used to determine the factors associated with ART adherence.

**Data Availability Statement:** Data cannot be shared publicly because of confidentiality issues in HIV care. However, the data is safely stored in

Muhimbili university and Hospital records with the reference number DA.287/298/01A/ Muhimbili University and Hospital Ethics commitee and will be available for researchers who meet the criteria for access to confidential data. The data underlying the results presented in the study are available under the Ref. No. D.A287/298/01A, Tel. No. +255-22-2150465, Email: dpgs@muhas.ac.tz, The Director of postgraduate studies, in Muhimbili University of Health and Allied Sciences in collaboration with director of Research and Publications, MUHAS.

**Funding:** The author(s) received no specific funding for this work.

**Competing interests:** The authors have declared that no competing interests exist.

**Abbreviations:** AIDS, Acquired Immunodeficiency syndrome; ART, Anti Retroviral Therapy; ARV, Anti Retroviral; CD4, Cluster of Differentiation 4; CTC, Care and Treatment Clinic; DPP, DARDAR Paediatric Program; HIV, Human Immunodeficiency Virus; IDC, Infectious Disease Centre; MNH, Muhimbili National Hospital; MUHAS, Muhimbili University of Health and Allied Sciences; PEPFAR, President's Emergency Plan for AIDS Relief; SPSS, Statistical Package of Social Science Software; SR, Recall-Self-Reported Recall; TRRH, Temeke Regional Referral Hospital; UNAIDS, Joint United Nations Programme on HIV and AIDS; VAS, Visual Analogue Scale.

Analysis was conducted using SPSS version 20.0 and p-value <0.05 were considered statistically significant.

## Results

333 participants were recruited. The overall good adherence ($\geq$95%) was approximated to be 60% (CI-54.3–65.1) when subjected to all three measures. On multivariable logistic regression, factors associated with higher odds of poor adherence were found to be caregivers aged 17–25 years [AOR = 3.5, 95%CI-(1.5–8.4)], children having an inter-current illness [AOR = 10.8, 95%CI-(2.3–50.4)], disbelief in ART effectiveness [AOR = 5.495; 95%CI-(1.669–18.182)] and advanced clinical stage [AOR = 1.972; 95% CI-(1.119–3.484)]. The major reasons reported by caregivers for missing medications included forgetfulness (41%), high pill burden (21%), busy schedule (11%) and long waiting hours at the clinic (9%).

## Conclusion and recommendations

In the urban setting of Dar es Salaam, ART adherence among children was found to be relatively low when combined adherence measures were used. Factors associated with poor ART adherence found were younger aged caregivers, and child intercurrent illness, while factors conferring good adherence were belief in ART effectiveness and lower HIV clinical stage. More attention and support should be given to younger aged caregivers, children with concomitant illness and advanced HIV clinical stages. Educating caregivers on ART effectiveness may also aid in improving adherence.

## Introduction

HIV (human immunodeficiency virus) is a virus that attacks the immune cells in the human body thus making them more vulnerable to infections and the development of a life-threatening chronic condition called AIDS (acquired immunodeficiency syndrome). To control HIV viral replication and improve the function of the immune system, patients must use ART (anti-retroviral therapy). Drug adherence is a key part of ART treatment, it refers to the whole process from choosing, starting, and managing to maintaining a given therapeutic medication regimen to control viral replication and improve the function of the immune system.

The HIV epidemic continues to have devastating consequences globally. According to UNAIDS, in 2018, 37.9 million people were living with HIV/AIDS worldwide, of which 1.7 million (5% of the population) were children aged less than 15 years [1]. Sub-Saharan Africa bears the largest burden of the HIV epidemic, accounting for 1.09 million children living with HIV/AIDS and more than 60% of new infections occur in women, infants or young children [2].

With the recent scale-up of ART access in sub-Saharan countries, the maximum focus needs to be laid on maintaining optimal adherence as it has been proven that, for the greatest success in ART effectiveness [3], adherence should be at 95% or greater [4–6]. The need for near-perfect adherence to a lifelong therapy from an early age has been identified as a major challenge in the administration of ART to HIV-infected children [4, 7, 8]. Poor adherence leads to low therapeutic drug levels, resulting in incomplete suppression of HIV replication.

It leads to resistance to the drugs and moreover cross-resistance within the same class [5, 6], thus compromising future treatment options and increasing mortality. Adherence reports among children taking ARTs in sub-Saharan Africa varies greatly 29–98% [9], with some countries reporting higher adherence over the other [10–13]. This can be attributed to the

diversity of methods being used to assess adherence where self-reported measures are the commonest employed amongst low resource settings and thus have a tendency to overestimate adherence compared to the objective methods.

It has been shown in these settings, that a more accurate assessment of ART adherence can be obtained when two or more methods are employed to assess adherence [14]. Studies on adherence amongst children in Tanzanian context are greatly lacking, a study done in 2012 revealed good adherence to ART regimen was only reported in 24.6% of children and adolescents when three assessment measures were employed.

Adherence has been seen to be complex amongst children as compared to adults with different pressing issues affecting it, as they have to rely on the caregivers and their wellbeing to obtain drugs and face multiple challenges in social, health system and drug-related aspects. Poor socio-economic circumstances including age of caregivers, poverty, single marital status and stigma were seen to be associated with poor adherence among countries like Cape Town, Uganda, Ethiopia, Nigeria and Northern Tanzania [10, 12, 15–17]. Various studies have related poor adherence to caregiver factors, as seen with children who are under the care of non-parental guardians and single parents or busy caregivers [18–21].

In some countries, the regimen and health system-related factors have been seen to pose a barrier for good adherence, whereby complexity, tediousness of the pediatric regimen were noted in some places like South India and Ethiopia [10, 22] and drug side effects and nausea were seen to impede adherence in Northern Tanzania and Jamaica [17, 19].

Enhancing adherence and understanding its barriers in the local context is crucial and must be an ongoing process to ensure ART success. Data amongst children and early teenagers is lacking in Tanzania and specifically in Dar es Salaam. This study aims to cater for this gap and assess the magnitude and attributes of ART adherence in this special group in order to design appropriate interventions to improve, or maintain optimal adherence levels.

## Materials and methods

### Study design and area

This was a hospital based cross-sectional study conducted between July to November 2018 in three sites within the city centre of Dar-es-Salaam; Muhimbili National Hospital (MNH), Temeke Regional Referral Centre (TRRH) and Infectious Disease Centre-DarDar Paediatric Clinic (IDC-DPP) HIV care and treatment clinics catering for children 1–14 years with their caregivers' and have monthly scheduled follow-ups' in accordance with the national guidelines for the management of HIV/AIDS. The three sites were chosen as they were more accessible and represented three different levels of health facilities in the country.

### Participants and eligibility criteria

The study participants were confirmed HIV positive children (1–14 years) already initiated on anti-retroviral therapy for at least six months and have scheduled follow up clinic visits with their caregivers at the three afore-mentioned CTC sites. Those on presumptive treatment were excluded. Presumptive treatment is the initiation of ARTs in an infant of an HIV positive mother with certain criteria before the availability of HIV PCR results. This is part of the Tanzania's national HIV guidelines [23].

We screened the child's clinical hospital records, CTC 1 and 2 cards for confirmation of the diagnosis of HIV and the duration of treatment and follow up at the CTC. Pharmacy records were checked for confirmation of attendance for drug pick up. Participants and their caregivers were consecutively enrolled until the sample size was achieved, if they fulfilled the inclusion criteria and consented.

The sample size for this study was calculated using the Kish Leslie formula using the prevalence of good adherence from the KCMC, a study in Northern Tanzania which employed three methods to assess adherence and found it to be 24.6% when all three methods were combined thus;

$$n = \frac{Z^2 \times P(1-P)}{ME^2}$$

Where,

Z—Is percentage point corresponding to a significance level of 5% (1.96)

P—Prior judgment of the correct level of ART adherence in children with HIV (%)

Taken as: **24.6% in a study done at KCMC between 2011–2012** [17]

E—Precision of the estimate = 0.05

Thus $n = \frac{1.96^2 * 0.246 * (1-0.246)}{0.05^2} = 285$

Therefore, 285 was the minimum sample size required for the study.

Considering 10% non-response, the required minimum sample size estimated was **316** participants with their caregivers.

To ensure fair distribution from each site, the sampling technique used was Proportion to size sampling, based on the number patients attending/registered at the specific CTC clinic. Then participants were consecutively recruited until the desired site contribution was reached and the final sample size was obtained in order to achieve a representative, yet, unbiased sample. The percent contribution towards the sample size was calculated as follows:

**MNH** had 255 patients 1–14 years registered thus contributing 35% of the sample size = **111** participants

**TRRH** had 250 patients 1–14 years registered thus contributing 34% = **107** participants

**IDC-DPP** clinic had 230 patients 1–14 years registered thus contributing 31% = **98** participants

In our study a total of 330 met the inclusion criteria during the recruitment process as described in Fig 1. The sample can be considered representative of a larger population as it included patients from three different cadres of the health facility.

## Variables and data measurement

The variables measured in this study included: adherence level, factors associated with ART adherence after being on ART for at least six months. ART adherence being the dependent variable and independent variables were socio-demographic characteristics like child's age, gender, schooling status, caregiver's age, relation to the child, marital status, employment status and monthly income. Participant characteristics like HIV status disclosure, stigmatization, child's illness, availability of adequate food, belief in ART effectiveness and HIV clinical stage, while treatment related factors considered in this study were type of ART regimen, duration of use, pill burden and side effects. Finally, health system related factors included in this study were drug availability, adequate refills, easy access to health facility and counseling on drug adherence.

**Measures of adherence.** Three measurement tools were used to assess adherences for all participants: four-day self-recall report, one-month self-recall report (using the visual analogue scale) and number of scheduled clinic appointments missed. For each of these measures, greater than or equal to 95% responses were considered as good adherence, while less than that were taken as poor adherence [24].

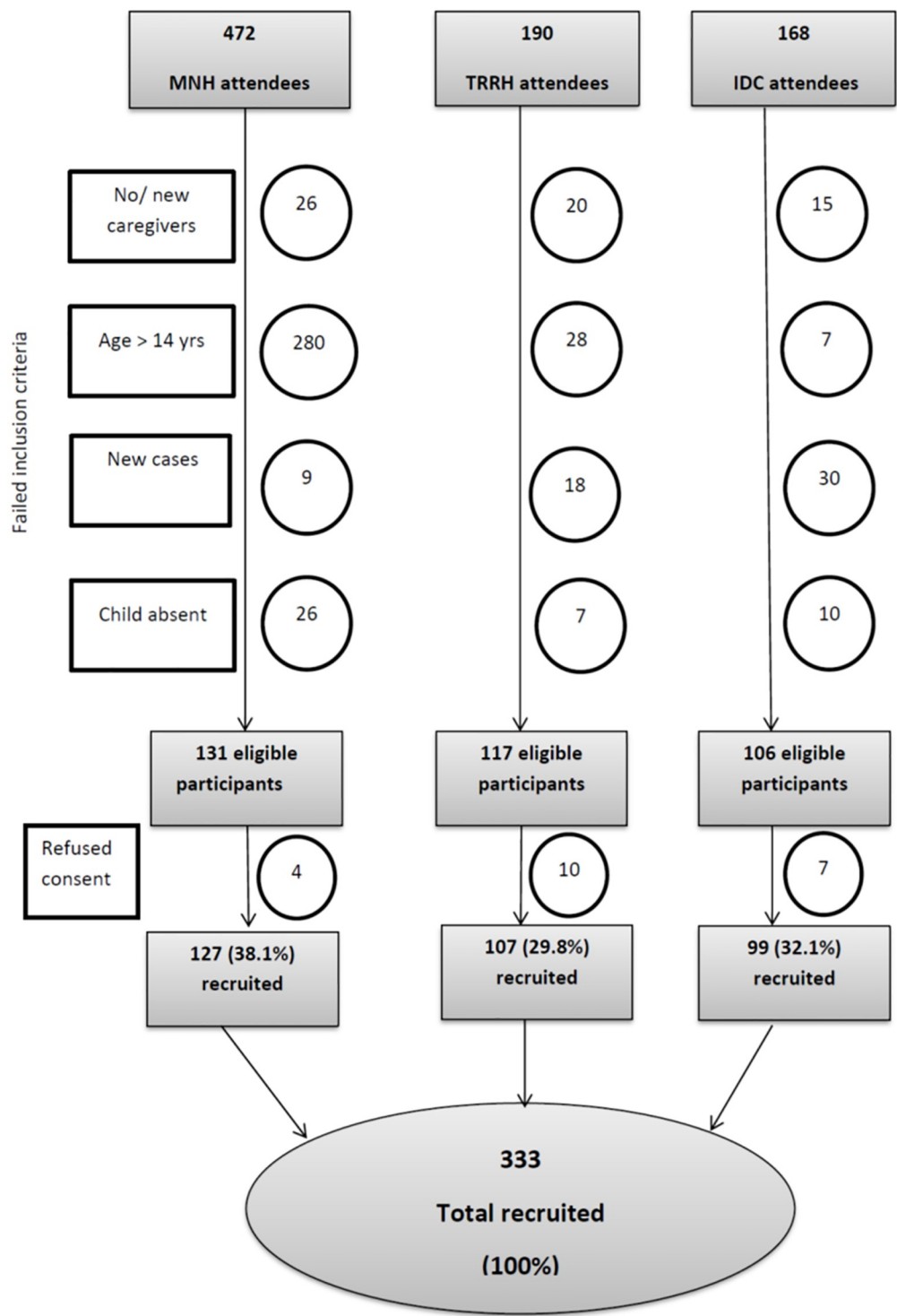

**Fig 1. Flow diagram showing recruitment of participants.**

4-day self-recall report: Caregivers were asked to recall if they had missed any dose over the last 4 days. If they reported missing more than one dose, it was considered as poor adherence <95%. If they reported one or no missed doses it was considered good adherence (≥95%).

1 month-self recall report (Visual Analogue Scale): Caregivers were asked to mark on a calibrated line scaled 1–10, as a reflection of the way they administered the medications to the child over the last one month. Thereafter, the mark was translated into percentage to estimate their drug administration estimate over the entire month. If the mark was below 9.5 (<95%) it was considered as poor adherence. If marked was at or above 9.5 (≥95%) it was taken as good adherence.

The number of missed clinic appointments: Records of the number of visits scheduled over the past 6 months were asked and confirmed from the CTC card and then the number of visits missed/ delayed according to the clinic's allowed time for delay, were checked from their CTC card and recorded accordingly. Each CTC had a maximum tolerance for delay where they were provided adequate drugs for that delay period. If there was no attendance beyond the allowed delay period, it was considered a missed visit. If the participant missed his/her scheduled appointment but attended it within the maximum allowed by the CTC, it was not considered a delay. This was further counter-checked with the pharmacies' refill records to ensure the drugs were picked up. If the participant missed more than one scheduled clinic visit it was considered as poor adherence (<95%).

Thus, after documenting all the three adherence measures, overall good adherence was qualified when the participant did well (≥95%) in all three adherence measures. If he/she performed less than 95% in any one adherence measure, he/she was qualified as having poor overall adherence.

**Factors affecting adherence.** Factors affecting ART adherence taken in this study were: socio-demographic factors, participant-related factors, drug regimen-related factors and health system-related factors, obtained from interviewing the caregiver.

**Data collection methods- tools and technique.** Data was collected using a researcher-administered, standardized structured questionnaire, which was developed in English and translated into Kiswahili, the national language in Tanzania to enable participants to respond in a language they fully comprehend. The interview was conducted with the caregiver of the participant, and the questionnaire was simultaneously filled during the interview. Data was collected by the principal investigator and a single research assistant who underwent a week of daily training by the principal investigator before the commencement of data collection. The research assistant was a medical officer who had good understanding about the subject of the study. Screening to identify the confirmed HIV positive children was conducted at the HIV CTC. The investigator first established a good rapport with the client and displayed a non-judgmental, empathic attitude before embarking on the interview, which was conducted in a private comfortable room with no interference/on sight listeners. Data collection forms were anonymous and were only identified by a unique study identification number. Some information was obtained from their medical records (CTC cards, case notes and online system) whichever was being used in that facility. For participants with indicators of poor adherence levels, reasons for poor adherence were enquired from the caregivers.

## Statistical analysis

Data was analyzed using SPSS software version 20. Quality control checks were assured during data entry process. Measures of central tendency were used to summarize discrete data. Charts, graphs and tables were used to display categorical data. The overall level of ART drug adherence was calculated, and contingency tables were constructed for bivariate analysis, to explore factors

associated with levels of ART adherence. Odds ratio was used to determine associations between the dependent and independent variables. Level of significant association was set at $p<0.05$.

Bivariate analysis and multivariate logistic regression models were used to determine Odds ratios and p values for varied factors associated with adherence. Only those factors whose Odds ratios had p values<0.2 on Bivariate analysis were further adjusted for in grouped manner on multivariate analysis. Adjusted Odds ratios with p values<0.05 on multivariate analysis were considered significant.

## Ethical consideration

Ethical approval to conduct the study was obtained from the Ethics Review Committee of Muhimbili University of Health and Allied Sciences and permission was sought from the respective health facilities. A written informed consent was sought from the caregiver of the child after providing them clear information regarding the study, its benefits, and risks of participating in this study.

A formal written assent was obtained from all children seven years and above who had the capacity to understand, after explaining to them in a simple language the aim of the study and the necessary information at a level they could comprehend.

Confidentiality of the participants' information was ensured throughout, and no identification was used on the data collection tools. A unique case report form (CRF) identification number was used on the questionnaire, such that no information could be traced back to the participants. Information obtained was stored in a password-secured computer database and the hard copies of the questionnaire were secured in a locked cabinet by the principal investigator.

## Results

### Socio-demographic characteristics of the study participants

Among the 333 participants as shown in Table 1, the median age was 11 years with an interquartile range of 7–13 years. A slightly higher proportion of the participants were male (53.2%) and more of the participants were in the adolescent age group (10-14yrs) (61.3%), and the majority were attending school (90.7%).

Most of the participants (82.9%), had at least one parent and for most, they were their biological parent (73.6%). About one-third of caregivers (34.5%) were unemployed and the majority of caregivers (75%) had an average gross monthly income of under 100,000 Tzs. Caregivers were aged between 17 and 75 with a median age of 40 years (IQR 34-46yrs), of which 70% were above 35 years of age.

### ART Adherence among HIV infected children

Amongst a total of 333 participants that were recruited, approximately 60% (CI- 54.3–65.1) were noted to have good adherence as shown in Fig 2.

Fig 3 displays the ART adherence level by each method of adherence assessment.

The four-day self-recall and missed clinic visits had an almost similar adherence rate of 93–94% as compared to the one-month self-recall report using VAS which showed a considerably lower level of adherence (61.6%). However, this was concordant with the overall adherence level.

### Factors associated with ART adherence

**Socio-demographics factors associated with ART adherence.** Table 2 shows the univariate analysis of the selected socio-demographic characteristics in relation to ART adherence. The only factor noted to be associated with ART adherence was the caregivers' age group.

**Table 1. Participant and caregiver socio-demographic characteristics.**

| Variable | Frequency (%) n = 333 |
|---|---|
| **Sex of Child** | |
| Male | 177 (53.2) |
| Female | 156 (46.8) |
| **Child Age Group (years)** | |
| Pre-Adolescent (1–9) | 129 (38.7) |
| Adolescent(10–14) | 204 (61.3) |
| **Child Schooling Status** | |
| Attending school | 302 (90.7) |
| Not attending school | 31 (9.3) |
| **Parental Status** | |
| Single/Both Parents | 276 (82.9) |
| Non-parental caregiver | 57 (17.1) |
| **Relation of Caregiver** | |
| Biological Parent | 245 (73.6) |
| Non-Biological Parent | 88 (26.4) |
| **Caregiver Age Group** | |
| 17-<25 years | 26 (7.8) |
| 25–<35 years | 74 (22.2) |
| > 35 years | 233 (70.0) |
| **Caregiver Marital Status** | |
| Married/Cohabiting | 177 (53.2) |
| Single/Divorced/Widowed | 156 (46.8) |
| **Employment Status** | |
| Employed | 218 (65.5) |
| Unemployed | 115 (34.5) |
| **Monthly Income** | |
| Less than 100,000 | 250 (75.1) |
| 100000 and above | 83 (24.9) |

Children whose caregivers were 17–25 years had a two times higher odds of having poor adherence [OR = 2.636, 95% CI 1.146–6.066 p = 0.023] when compared to children whose caregivers were over 25 years.

On further investigation about these younger aged caregivers, it was noted six amongst the twenty-six were older siblings caring for their younger ones in the absence of their parents and the remaining were young mothers.

**Participant factors associated with ART adherence.** Participant factors that were associated with poor ART adherence on univariate analysis, were found to be child having an inter-current illness had eighteen times higher odds of having poor adherence (OR = 18.3; 95% CI 4.214–79.513; p<0.001) and those who had adequate access to food, were two and a half times more likely to be associated with poor adherence (OR = 2.507; 95% CI 1.452–4.330; p = 0.001). In addition, it was seen in this study that disbelief in ART effectiveness and advanced clinical stage was associated with higher odds of poor adherence (OR = 8.474; 95%CI 3.145–22.727; p<0.001) (OR = 1.927; 95% CI 1.149–3.23; p = 0.019) respectively and these were found to be statistically significant as seen in Table 3.

**Regimen related and health system-related factors associated with ART adherence.** Tables 4 and 5 demonstrate the selected regimen related and health facility factors assessed in this study. None of these factors were significantly associated with poor adherence.

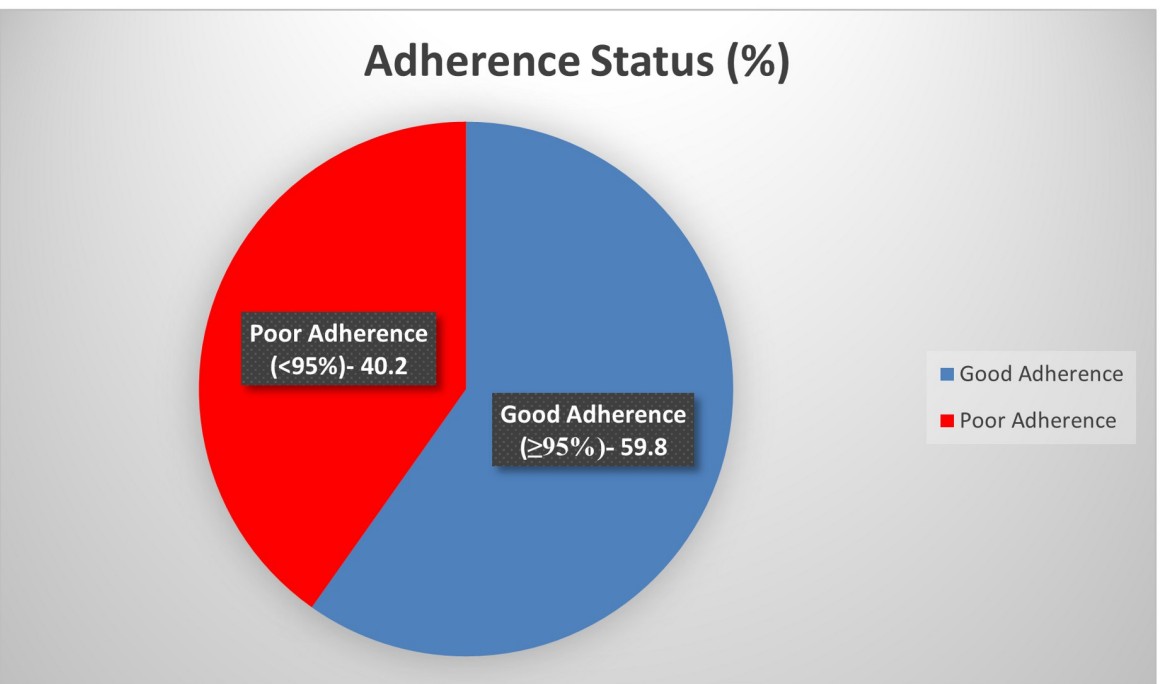

**Fig 2. Overall level of ART adherence among children 1–14 years.**

**Independent factors associated with poor ART adherence.** The possible factors that were seen to be associated with ART adherence on univariate analysis were further analysed for confounder effect. The Hosmer-Lemshow goodness of fit test for logistic regression was used to test the association, the test statistic equalled a p-value of 0.243, which signified that the

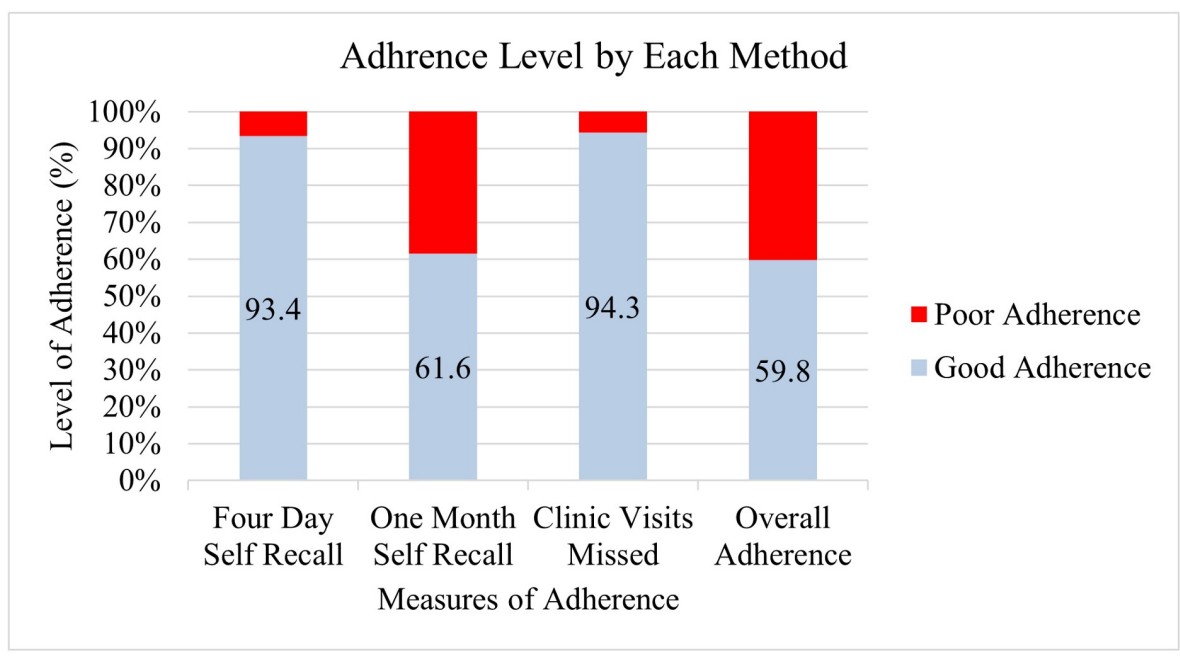

**Fig 3. Adherence level By each method of adherence assessment.**

**Table 2. Socio-demographic characteristics associated with ART adherence.**

| Variable | Adherence | | Total (n = 333) | OR (95% CI) | P-Value |
|---|---|---|---|---|---|
| | Poor (%) | Good (%) | | | |
| **Gender** | **(n = 134)** | **(n = 199)** | | | |
| Male | 74(41.8) | 103(58.2) | 177 | 1.150 (0.741–1.784) | 0.534 |
| Female | 60(38.5) | 96(61.5) | 156 | | |
| **Child Age Group** | | | | | |
| Pre-Adolescent (1-9yrs) | 45(34.9) | 84(65.1) | 129 | 0.692(0.439–1.092) | 0.113 |
| Adolescent (10-14yrs) | 89(43.6) | 115(56.4) | 204 | | |
| **Child Schooling Status** | | | | | |
| Attending school | 121(40.1) | 181(59.9) | 302 | 0.926(0.437–1.959) | 0.840 |
| Not attending school | 13(41.9) | 18(58.1) | 31 | | |
| **Parental Status** | | | | | |
| Single/Both Parents | 110(39.0) | 166(60.1) | 276 | 0.911 (0.511–1.625) | 0.762 |
| Non-parental caregiver | 24(42.1) | 33(57.9) | 57 | | |
| **Relation of Caregiver** | | | | | |
| Biological Parent | 95(38.8) | 150(61.2) | 245 | 0.796(0.486–1.303) | 0.364 |
| Non-Biological Parent | 39(44.3) | 49(55.7) | 88 | | |
| **Caregiver Age Group** | | | | | |
| 17–<25 years | 16(61.5) | 10(38.5) | 26 | 2.636 (1.146–6.066) | **0.023** |
| 25–<35 years | 30(40.5) | 44(59.5) | 74 | 1.123 (0.658–1.917) | 0.669 |
| > 35 years | 88(37.8) | 145(62.2) | 233 | 1 *Reference variable* | |
| **Caregiver Marital Status** | | | | | |
| Married/Cohabiting | 71(40.1) | 106(59.9) | 177 | 0.989 (0.638–1.534) | 0.960 |
| Single/Divorced/Widowed | 63(40.4) | 93(59.6) | 156 | | |
| **Employment Status** | | | | | |
| Employed | 86(39.4) | 132(60.6) | 218 | 0.909 (0.574–1.440) | 0.685 |
| Unemployed | 48(41.7) | 67(58.3) | 115 | | |
| **Monthly Income** | | | | | |
| Less than 100,000 | 101(40.4) | 149(59.6) | 250 | 1.027(0.619–1.705) | 0.918 |
| 100000 and above | 33(39.8) | 50(60.2) | 83 | | |

logistic regression model used to analyse this data was appropriate. Multivariate analysis results for those factors and additionally others with p<0.2 are displayed in Table 6. It was seen that the factors which remained to be significantly associated with a higher odds of poor adherence, were younger age group caregivers 17–25 years (AOR = 3.520; 95% CI 1.471–8.422), children having inter-current illness (AOR = 10.869; 95% CI 2.340–50.489), disbelief in ART effectiveness (AOR-5.495; 95% CI-1.669–18.182) and advanced children clinical stage (AOR-1.972; 95% CI-1.119–3.484).

Availability of adequate food lost statistical significance on multivariate analysis. Moreover, Child age group, HIV disclosure, clinic timings and counselling were not found to be associated with poor ART adherence.

## Caregiver reasons for missing doses in HIV infected children

For those caregivers who reported missing even one pill, reasons were explored and some of the main challenges cited by them were; forgetting to take pills daily (41%) and high pill burden (21%) as the predominating factors followed by busy schedule- interrupting drug administration (11%) and long waiting hours at the clinic (9%). These, among the other cited reasons, are shown in Fig 4.

**Table 3. Participant characteristics associated with ART adherence.**

| Variable | Adherence | | Total (n = 333) | OR (95% CI) | P-Value |
|---|---|---|---|---|---|
| | Poor (%) | Good (%) | | | |
| **HIV Status disclosure** | (n = 134) | (n = 199) | | | |
| Yes | 66(45.2) | 80(54.6) | 146 | 1.444(0.928–2.245) | 0.103 |
| No | 68(36.4) | 119(63.6) | 187 | | |
| **Experiencing Stigma** | | | | | |
| Yes | 121(40.5) | 178(59.5) | 299 | 1.098(0.530–2.277) | 0.801 |
| No | 13(38.2) | 21(61.8) | 34 | | |
| **Child Inter-current illness** | | | | | |
| Yes | 21(91.3) | 2(8.7) | 23 | 18.3(4.214–79.513) | <**0.001** |
| No | 113(36.5) | 197(63.5) | 310 | | |
| **Availability of Adequate Food** | | | | | |
| Yes | 39(58.2) | 28(41.8) | 67 | 2.507(1.452–4.330) | **0.001** |
| No | 95(35.7) | 171(64.3) | 266 | | |
| **Believf in ART Effectiveness** | | | | | |
| Yes | 110(36.2) | 194(63.8) | 304 | | <**0.001** |
| No | 24(82.8) | 5(17.2) | 29 | 8.474(3.145–22.727) | |
| **Clinical Stage** | | | | | |
| Stage 1–2 | 94(36.6) | 163(63.4) | 257 | | **0.012** |
| Stage 3–4 | 40(52.6) | 36(47.4) | 76 | 1.927(1.149–3.23) | |

## Discussion

ART adherence in children is crucial to ensure treatment success. Poor adherence results in sub-optimal drug levels making it less effective in suppressing viral replication resulting in treatment failure. There is an emerging concern as to what level of adherence can be achieved

**Table 4. Drug regimen related characteristics associated with ART adherence.**

| Variable | Adherence | | Total (n = 333) | OR (95% CI) | P-Value |
|---|---|---|---|---|---|
| | Poor (%) | Good (%) | | | |
| **Type of ART Regimen** | (n = 134) | (n = 199) | | | |
| Lopinavir/ritonavir based | 27(37.5) | 45(62.5) | 72 | *1 Reference Variable* | |
| Efavirenz based | 26(32.5) | 54(67.5) | 80 | 0.802(0.411–1.565) | 0.519 |
| Nevirapine based | 81(44.2) | 100(55.2) | 181 | 1.350(0.771–2.363) | 0.293 |
| **ART Duration Use** | | | | | |
| Less than 2 years | 12(36.4) | 21(63.6) | 33 | 0.834(0.396–1.757) | 0.633 |
| 2 years and more | 122(40.7) | 178(59.3) | 300 | | |
| **Number of pills per dose** | | | | | |
| 1-pill | 30(45.5) | 36(54.5) | 66 | 1.306 (0.759–2.249) | 0.335 |
| More than 1 pill | 104(39.0) | 163(61.0) | 267 | | |
| **Person administering dose** | | | | | |
| Child him/herself only | 10(55.6) | 8(44.4) | 18 | 1.846(0.692–4.927) | 0.221 |
| Caregiver only | 59(38.3) | 95(61.7) | 154 | 0.917(0.583–1.442) | 0.708 |
| Child and caregiver | 65(40.4) | 96(59.6) | 161 | *1 Reference Variable* | |
| **Side effects Experienced** | | | | | |
| Yes | 19(48.7) | 20(51.3) | 39 | 1.479(0.757–2.890) | 0.253 |
| No | 115(39.1) | 179(60.9) | 294 | | |

**Table 5. Health system-related characteristics associated with ART adherence.**

| Variable | Adherence | | Total (n = 333) | OR (95% CI) | P-Value |
|---|---|---|---|---|---|
| | Poor (%) | Good (%) | | | |
| **Drug Change due to Unavailability** | **(n = 134)** | **(n = 199)** | | | |
| Yes | 39(39.8) | 59(60.2) | 98 | 0.974(0.602–1.576) | 0.915 |
| No | 95(40.4) | 140(59.6) | 235 | | |
| **Adequate Drug Refill** | | | | | |
| Yes | 107(41.5) | 151(58.5) | 258 | 1.260(0.740–2.146) | 0.395 |
| No | 27(36.0) | 48(64.0) | 75 | | |
| **Difficulty Accessing Health Facility** | | | | | |
| Yes | 76(41.8) | 106(58.2) | 182 | 1.15(0.740–1.787) | 0.535 |
| No | 58(38.4) | 93(61.6) | 151 | | |
| **Clinic Time Convenient** | | | | | |
| Yes | 57(35.0) | 106(65.0) | 163 | 0.649(0.418–1.010) | 0.055 |
| No | 77(45.3) | 93(54.7) | 170 | | |
| **Counseled on importance of Adherence** | | | | | |
| Yes | 130(39.6) | 198(60.4) | 328 | 0.164(0.018–1.485) | 0.108 |
| No | 4(80.0) | 1(20.0) | 5 | | |

from children in poor resource settings, since adherence in children and adolescents has been identified as a challenge due to multiple factors, dependence on others to give them the drugs and problems related to the endurance of life-long therapy.

The overall ART adherence of approximately 60%, is in keeping with a study done in Nigeria in 2015, and close to that reported in rural Tanzania, during the same year 65.6% and 70% respectively [16, 20]. A systematic review of ART adherence in low and middle-income countries in 2008 found a range in adherence estimates from 49% to 100% using different adherence measures [25].

In contrast to our findings, adherence was reported to be higher ranging from 72% to 94% among the last two decades in other sub-Saharan countries including; Ethiopia, Tanzania, Malawi and Uganda, with most studies reporting>90% ART adherence by self-recall means [10–13]. Similarly, a single centred study done in Northern Tanzania, subjected to the three methods, reported a significantly lower adherence level of 24.6% among HIV infected children [17]. Our comparatively higher adherence estimates despite multiple methods, could be explained by the considerable improvement in CTC services, free ART access and improved adherence counselling over the past half-decade.

The possible explanation of the exceptionally high contrasting results, obtained by the other sub-Saharan countries, could be due to their use of short-term recall methods to assess adherence. In fact, in this study, using the four-day recall method alone to measure adherence would have shown similar high adherence of 93.4% as these studies. This method, when used alone, is known to over-estimate ART adherence, especially if caregivers are the ones reporting, due to their vulnerability to social desirability and over-reporting. These biases could result in falsely inflated adherence estimates in the actual situation [25]. While the VAS score is subjected to recall bias, it is the only measure which may capture the participant's own perception of his/her adherence, a measure of self-evaluation and may actually affect the outcome of future adherence and thus is an important measure that should not be omitted.

Multivariate logistic regression analysis indicated that the age of caregiver was an independent factor associated with ART adherence. Children, whose caregivers were 17–25 years, were significantly found to be associated with poor adherence. These findings were similar to

**Table 6. Independent factors associated with poor ART adherence.**

| Variables | Adherence | | cOR (95%CI) | AOR (95% CI) | P-Value |
|---|---|---|---|---|---|
| | Poor(%) | Good(%) | | | |
| **Child Age Group** | | | | | |
| Pre-Adolescent (1-9yrs) | 45(34.9) | 84(65.1) | 0.692(0.439–1.092) | 0.848 (0.453–1.587) | 0.607 |
| Adolescent (10-14yrs) | 89(43.6) | 115(56.4) | | 1 | |
| **Care Giver Age Group** | | | | | |
| 17-<25 years | 16(61.5) | 10(38.5) | 2.636(1.146–6.066) | 3.520 (1.471–8.422) | **0.005** |
| 25–<35 years | 30(40.5) | 44(59.5) | 1.123(0.658–1.917) | 1.431 (0.781–2.619) | 0.246 |
| > 35 years | 88(37.8) | 145(62.2) | 1 | 1 | |
| **HIV Status Disclosure** | | | | | |
| Yes | 66(45.2) | 80(54.6) | 1.444(0.928–2.245) | 1.303 (0.717–2.367) | 0.385 |
| No | 68(36.4) | 119(63.6) | | 1 | |
| **Child inter-current Illness** | | | | | |
| Yes | 21(91.3) | 2(8.7) | 18.3(4.214–79.513) | 10.869 (2.340–50.489) | **0.002** |
| No | 113(36.5) | 197(63.5) | | 1 | |
| **Availability of Adequate Food** | | | | | |
| Yes | 39(58.2) | 28(41.8) | 2.507(1.452–4.330) | 1.574 (0.830–2.984) | 0.165 |
| No | 95(35.7) | 171(64.3) | | 1 | |
| **Believes on Effectiveness of ART** | | | | | |
| Yes | 110(36.2) | 194(63.8) | | 1 | |
| No | 24(82.8) | 5(17.2) | 8.474(3.145–22.727) | 5.495(1.669–18.182) | **0.005** |
| **Clinical Stage** | | | | | |
| Stage 1–2 | 94(36.6) | 163(63.4) | | 1 | |
| Stage 3–4 | 40(52.6) | 36(47.4) | 1.927(1.149–3.23) | 1.972(1.119–3.484) | **0.019** |
| **Clinic Timing Convenient** | | | | | |
| Yes | 57(35.0) | 106(65.0) | 0.649(0.418–1.010) | 0.711 (0.430–1.177) | 0.185 |
| No | 77(45.3) | 93(54.7) | | 1 | |
| **Counseled on Importance of Adherence** | | | | | |
| Yes | | | 0.164(0.018–1.485) | 1.448 (0.106–19.706) | 0.781 |
| No | | | | 1 | |

those reported from Kano (Nigeria) and Mekelle (Ethiopia) [10, 16]. However, caregivers' marital status and their economic condition did not seem to have an effect in this study like it did in the aforementioned surveys. Similarly, findings from KCMC in Northern Tanzania, suggested caregivers' low monthly income was associated with poorer adherence [17]. The caregivers' relationship to the child had little effect on adherence status of the child, unlike the findings reported from a region in rural part of Tanzania-Ifakara in 2015 by Nyogea *et al* from Tanzania where step mothers contributed to poor adherence [20].

A child's optimal adherence strongly relies on the caregiver's commitment; the younger age group are more likely to be less responsible towards this entrustment than their counterparts, who may have a better understanding of the dynamics of disease and health and thus, deserve more focused attention on adherence counselling and may entail health providers to identify older household members to be involved hand in hand in the care of the child.

## Regimen related and health facility related factors affecting adherence

The study took place in centres with trained and motivated staff, where all the facilities provided care which was in line with the national ART scale up policy, by using simple once/twice

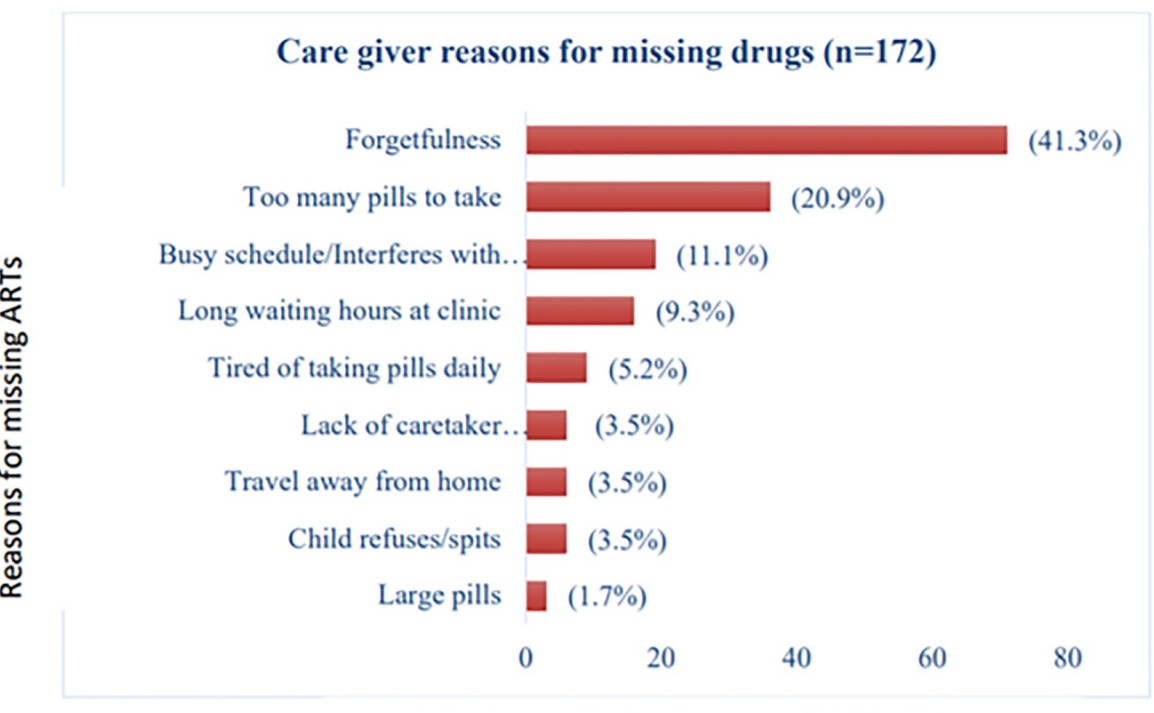

**Fig 4. Reasons for missing doses in HIV infected children.**

daily regimens, fixed dose combinations when available and constant supply at all times. Due to the fully funded HIV CTC services, participants no longer faced the regimen related and healthcare related barriers to ART adherence including: cost of drugs, difficulty in accessing healthcare facilities, inadequate refills, unqualified regimen changes and pill burden. Therefore, these factors were not found to significantly influence ART adherence in this study; which was in line with other studies in Sub-Saharan African countries with similar setups in providing ART services [13, 17, 26–28].

This study highlighted that, the most significant barriers to good ART adherence were those related to the participant themselves; Child's intercurrent illness, availability of adequate food, belief in the effectiveness of ARTs and HIV clinical stage. In a few studies, some but not all these factors were found to be obstacles for ART. A study done in Nigeria in 2010, reported child's inter-current illness and non-belief in ARTs; while a study in rural Tanzania (Ifakara) found preferring herbal medicine and experiencing stigma as barriers for ART adherence [16, 20]. On the contrary, a study in urban Malawi, Ethiopia and northern Tanzania found no significant relation of ART adherence with a child's WHO clinical stage or caregivers' belief in ART effectiveness [13, 17, 29]. Non-disclosure and stigma were not found to affect adherence in this study as it did in earlier studies [12, 30]. The most plausible reason for this difference may be due to improved awareness, acceptance of HIV/AIDS and encouragement of timely child disclosure.

To explain our findings, we speculate; in Tanzania, while the knowledge of HIV/AIDS and its treatment is overall good, beliefs of an existing supernatural power that may have afflicted an HIV infected person still exists, leading to a preference for alternative healing (herbs, spiritual medicine) which may hinder ART use. The findings point to a possible serious gap in a caregiver's awareness of health implications of non-adherence during illness and advanced

stage, coupled with a false belief that taking ARVs with other drugs or during concomitant illness could result in serious adversities. Further studies on caregivers' belief aspects need to be carried out to confirm this finding.

Surprisingly in this study, the availability of adequate food was found to be associated with poor adherence, similar to findings from Addis Ababa [29], however this lost significance on multivariate analysis. It is difficult to speculate about this association, as several studies have deemed nutrition availability and support as an integral part of successful HIV care and treatment [31–33] losing its significance on further analysis verifying its confounder effect. Moreover, under-nutrition was noted among less than one-fifth of the participants (17%) signifying that it poses less of a problem in recent times than it did around a decade earlier in Tanzania [27] and Ethiopia [26]. The possible explanation for this could be that most of these patients were not ART naïve and had been attending the clinic and receiving counseling, which may have improved their nutritional status. Furthermore, nutrition is not entirely dependent on adherence to ARVs, rather is determined by multiple factors like educational and socioeconomic status, family size, dietary diversity and other comorbid conditions [33–35].

### Caregivers' reasons for missing medications

The main problems cited by the caregivers, who were responsible for a missed dose in this study were: forgetfulness, high pill burden, a busy schedule, and long waiting hours at the clinic. Some, but not all of these factors, were reported in other studies from similar resource limited countries like Ethiopia-Mekelle which found the pill a burdensome and a survey in Addis Ababa found forgetfulness (23.5%) [26] as barriers to observing adherence. A study done in Nigeria also reported similar findings as ours; forgetfulness (59.5%), travel away from home/busy (21.4%) and child inter-current illness (14.3%) [16].

A multi-centre study in Tanzania reported long waiting hours at the clinic to be a hindrance to effective adherence [27] which was similar to findings in this study. This is possible because some HIV clinics operated only once a week for children. In addition to that, with the current provision of free drugs, the demand is higher and more prone to having longer waiting hours, unlike previous times where patients had to pay for their drugs. This can lead to patient dissatisfaction and default from treatment [36]. The pill burden was observed to be high among most of the participants of this study and this could be due to the deficiency of paediatric fixed-dose combinations available for every ART regime, as is available for adults- as a single pill taken once a day.

On the contrary, earlier studies reported explanations of suboptimal adherence being the cost of transportation, difficulty accessing health facilities, cost of medication and lack of adequate counselling [27, 37]. During those times, HIV care was not entirely free and widely distributed and accessible. Patients had to bear some costs of their treatment, including buying their drugs and paying for lab tests. Currently, with support funding from PEPFAR, HIV drugs and services is cost-free, and clinics and transport services are easily accessed and thus not reported to be problematic.

The strengths of this study were that it was conducted in three centres in the region including primary and tertiary level health facilities, serving participants from most districts. Therefore, the data obtained is reflective of the actual situation of adherence in our surroundings. Furthermore, the parameters used to assess adherence were practical and easily employed in any low-resource clinical setting. Three separate measures were in cooperated, to improve the reliability of the assessment, and thus remove the bias of one measure over the other.

This study should be interpreted with the following limitations in mind: Cross-sectional studies of this nature are unable to capture trends in adherence, which are known to change

over time, thus are limited in drawing conclusions regarding causal effect relationship of potential factors, a longitudinal component may be the most appropriate for this regard. In addition, classification of different adherence cut-off points, adherence measures and study designs used in different studies may not be perfect to compare and contrast findings meticulously. With regards to the parameters used to assess adherence, they are liable to be affected by recall bias and social desirability bias and these may have over-estimated good adherence. However, efforts were made to mitigate them by employing three different methods, whereby, the 4-day recall method has nearly no recall bias and the clinic attendance method is highly objective and concrete. The one-month recall method is actually the most accurate depiction of the care givers' own perception of the child's adherence and has implications on his/her future adherence pattern and is thus an important parameter that should not be excluded.

In conclusion, good adherence was noted only among 60% of the children attending HIV CTC clinics in Dar es Salaam. The factors influencing poor adherence were; younger age of caregivers and a child having inter-current illness whilst, belief in the effectiveness of ARTs and a child in the initial HIV clinical stages fostered good adherence. The most cited caregiver-related reasons for missing drugs were; forgetfulness, high pill burden, busy schedule and long waiting hours at clinic.

Thus, we recommend the following; health workers should assess for ART adherence and routinely do adherence counseling at each encounter with the caregiver, whereby these simple adherence tools should be in-cooperated, in order to identify and detect poor adherence with more focus on child-centered adherence counseling strategies so as to further improve adherence in children in order to achieve 90-90-90 target.

Healthcare providers should also seek to enlist older caregivers above the age of 25 years in the household as treatment supporters for each child. They may team up with the younger caregivers to enhance ART adherence among children. Sick children and those with advanced HIV stages may need to be admitted or have a much closer follow-up with the clinician to improve their clinical status and adherence to treatment, as it may be demotivating to the caregiver when they see their child not improving giving them a reason to default treatment. It is during these periods when children most need their drugs to overcome their illness and improve their clinical stage. Furthermore, health education at point of service, like during the daily morning talks and on social media platforms about the effectiveness of ARVs' and alleviating the mythical beliefs about the HIV/AIDS and its treatment in order to improve adherence.

## Supporting information

**S1 File. The questionnaire.**
(DOCX)

## Acknowledgments

We would like to give special thanks to the caregivers and their children with HIV/AIDS who participated in this research, the administrative, medical and nursing staff of the Muhimbili National hospital, IDC- DarDar Paediatric Program HIV Clinic and Temeke Regional Referral Hospital. This work would have never been possible without them. We would also like to thank Dr Mucho and Candida Moshiro for their advice in the statistical aspects of this study. We are grateful to Mr MohammedHassan Lakha and Hussein Sajjad Hussein for their contribution in this manuscript.

## Author Contributions

**Conceptualization:** Fatima M. Mussa.

**Data curation:** Fatima M. Mussa.

**Formal analysis:** Fatima M. Mussa, Sibtain Moledina.

**Funding acquisition:** Fatima M. Mussa.

**Investigation:** Fatima M. Mussa.

**Methodology:** Fatima M. Mussa.

**Project administration:** Fatima M. Mussa.

**Resources:** Fatima M. Mussa, Hussein Bhalloo.

**Software:** Fatima M. Mussa.

**Supervision:** Higgins P. Massawe, Evelyne Assenga.

**Validation:** Fatima M. Mussa, Evelyne Assenga.

**Visualization:** Fatima M. Mussa, Evelyne Assenga.

**Writing – original draft:** Fatima M. Mussa.

**Writing – review & editing:** Fatima M. Mussa, Higgins P. Massawe, Hussein Bhalloo, Evelyne Assenga.

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
