## [Decision Letter · Decision Letter 0]

31 Dec 2020

PONE-D-20-25583

MAGNITUDE AND ATTRIBUTES OF ANTI-RETROVIRAL THERAPY ADHERENCE AMONG CHILDREN (1-14 YEARS) ATTENDING HIV CARE AND TREATMENT CLINICS IN DAR ES SALAAM, TANZANIA

PLOS ONE

Dear Dr. Mussa,

Thank you for submitting your manuscript to PLOS ONE. After careful consideration, we feel that it has merit but does not fully meet PLOS ONE’s publication criteria as it currently stands. Therefore, we invite you to submit a revised version of the manuscript that addresses the points raised during the review process.

We look forward to receiving your revised manuscript.

Kind regards,

Satya Surbhi, PhD

Academic Editor

PLOS ONE

Journal Requirements:

2. Please provide additional details regarding participant consent. In the ethics statement in the Methods and online submission information, please ensure that you have specified whether consent was informed.

3. Please provide your sample size calculation.

4. In your Methods section, please provide additional information about the participant recruitment method and the demographic details of your participants. Please ensure you have provided sufficient details to replicate the analyses such as:

a) the recruitment date range (month and year),

b) a statement as to whether your sample can be considered representative of a larger population, and

c) a description of how participants were recruited.

5. Please include a copy of the interview guide used in the study to interview caregivers, in both the original language and English, as Supporting Information, or include a citation if it has been published previously.

6. We note that you have indicated that data from this study are available upon request. PLOS only allows data to be available upon request if there are legal or ethical restrictions on sharing data publicly. For information on unacceptable data access restrictions, please see http://journals.plos.org/plosone/s/data-availability#loc-unacceptable-data-access-restrictions.

7. Thank you for stating the following financial disclosure:

8. Please amend either the title on the online submission form (via Edit Submission) or the title in the manuscript so that they are identical.

9. Your ethics statement should only appear in the Methods section of your manuscript. If your ethics statement is written in any section besides the Methods, please move it to the Methods section and delete it from any other section. Please ensure that your ethics statement is included in your manuscript, as the ethics statement entered into the online submission form will not be published alongside your manuscript.

10. We note you have included a table to which you do not refer in the text of your manuscript. Please ensure that you refer to Table 1 in your text; if accepted, production will need this reference to link the reader to the Table.

Reviewers' comments:

Reviewer's Responses to Questions

**Comments to the Author**

1. Is the manuscript technically sound, and do the data support the conclusions?

Reviewer #1: Yes

Reviewer #2: Yes

Reviewer #3: Yes

2. Has the statistical analysis been performed appropriately and rigorously? 

Reviewer #1: Yes

Reviewer #2: Yes

Reviewer #3: No

3. Have the authors made all data underlying the findings in their manuscript fully available?

Reviewer #1: Yes

Reviewer #2: Yes

Reviewer #3: No

4. Is the manuscript presented in an intelligible fashion and written in standard English?

Reviewer #1: No

Reviewer #2: Yes

Reviewer #3: No

5. Review Comments to the Author

Reviewer #1: The manuscript needs revision especially on data analysis, methods and discussion section. Besides this the ascent should be stated in ethical clearance for children from their parents or guardian. The factors that are insignificant should be deleted.

Reviewer #2: The manuscript is well-prepared. It is scientifically sound. It will be more clear and precise if the authors include the analysis tools in the methodology section of the abstract, give more background information on the pediatric ART programme in the country like the regimens introduced etc. In addition, even if the number of references are enough better to include more recent related publications. The paper cites references dated back to 2000 to 2018.

Reviewer #3: First, I would like to congratulate the authors for exploring such an important topic. Magnitude and Attributes Of Anti-Retroviral Therapy Adherence Among Children (1-14 Years) Attending HIV Care And Treatment Clinics In Dar Es Salaam, Tanzania. However I have some comments and questions for them, these are found below

Abstract:

Introduction:

Line 1: In Epidemiological disease level classification HIV is a Pandemic disease; use this term instead of Epidemic. In addition at the end of the introduction please add justification/the need to conduct this study in a line or two.

Objectives: Correct the English (Among HIV positive children). Objective must be SMART which includes the “T” which stands for time bound. Therefore specify the time of the study.

Methods: The first statement is not even a full sentence. The method presented is too shallow. The sample size, the method of data analysis, level of significance, the measurement of association, the model fitness assessment, the software used all are lacking.

Result: The magnitude of good adherence reported (60%) is a point estimate, you should provide the confidence interval. Take the type of analysis used in the method section, use the term multivariable logistic regression instead of multivariate (Multivariate is when you have multiple outcome variables). Age is better reported as a range instead of <25 for children of very young (underage) mothers may not have the same association. Belief in ART effectiveness is not clear (Negative or Positive?? And how did you assess it??). After all the associated factors were written the sentence was concluded as “”… were less likely to be associated poor adherence.” Yet all the significant variables are not negatively associated with the outcome variable. Besides, though, the authors reported the magnitude of good adherence, they reported associated factors for poor adherence, you need to be consistent.

Conclusion and recommendations: The authors should re-write this section. Most of it is wrong and the recommendations given are not based on the findings.

Methods and Materials

Study design and area: In the first line authors mentioned “This was hospital based descriptive cross-sectional study” which is totally wrong. You have assessed factors that affecting adherence therefore it is not mere description. (Use hospital based cross-sectional study)

Participants and eligibility criteria

The minimum sample size determination must be clearly presented. There is no such a thing as “quota sampling calculation” Quota sampling is a sampling technique (How you distribute the sample size you determined across quotas).

How was the three study places identified?? Why none probability sampling???

How can you take consent form under 14 years old children?? (This must be clearly described in the Ethical consideration)

Variables and measurements

Clearly put the outcome variable and list of independent variables considered.

Statistical analysis

What method of goodness of fit assessment did you use?? What was its value??

Generally the method part lacks the following: Study period, Sample size determination, Sampling Procedures, Data collection procedure and quality assurance.

Results

The first paragraph must be part of the method section. You should focus on what you have found and not how you did it in the result section.

The order of the result presentation is full of flaws. First report the characteristics of study participants be it socio-demographic characteristics or co-morbidities. Then go to the descriptive findings of your study which the level/magnitude of adherence, finally you can present the factors associated. The level/magnitude of adherence also needs to be reported using confidence intervals.

Factors associated with ART adherence

Why are the authors reporting the associated factors as socio-demographic and participant related and the like all variables must be adjusted in the multivariable analysis and be reported as one multivariable table.

6. PLOS authors have the option to publish the peer review history of their article (what does this mean?). If published, this will include your full peer review and any attached files.

Reviewer #1: **Yes: **Abebe Dilie Afenigus

Reviewer #2: **Yes: **Yesunesh Teshome

Reviewer #3: **Yes: **Adhanom Gebreegziabher Baraki

---

## [Author Response · Author response to Decision Letter 0]

9 Feb 2021

RESPONSES TO REVIEWERS

REVIEWER 1

Title: 

1. Better to say “associated factors” rather than “attributes” in the title.

RESPONSE: CHANGED TO ASSOCIATED FACTORS

2. Originality: what makes your study unique from a similar study conducted in your country which is available with the following link? 

https://bmcinfectdis.biomedcentral.com/articles/10.1186/s12879-015-0753-y?

RESPONSE: THIS STUDY WAS CONDUCTED IN A RURAL AREA IN IFAKARA IN 2015, IT WAS A MULTICENTRE STUDY WITH A SMALLER SAMPLE SIZE AND THAT MAY LIMIT GENERALIZABILITY EVEN IN OUR SETTING. A SINGLE MEASURE WAS USED TO ASSESS ADHERENCE. THEIR COHORT WAS A VERY WIDE AGE RANGE FROM 2-19 YEARS AND IT IS KNOWN THAT FACTORS AFFECTING ADHERENCE VARIES GREATLY AMONGST CHILDREN AND ADOLESCENTS.

ADHERENCE AND FACTORS AFFECTING IT, IS A DYNAMIC PROCESS THAT CHANGES OVER TIME AND NEEDS REGULAR ASSESMENT TO IMPROVE AND OVERCOME ITS BARRIERS, THAT WARRANTS US TO REVIEW THE OUTCOMES AS WE CHANGE AND IMPROVE OUR PRACTICES IN THE CARE OF THESE CHILDREN. FURTHERMORE, WE DID A MULTICENTRE STUDY, CHOSING DIFFERENT CADRES OF THE HEALTH SYSTEM FROM THE COUNTRYS NATIONAL REFERAL HOSPITAL TO A REGIONAL HOSPITAL CENTRE, TO A PRIMARY LEVEL OUT PATIENT CLINIC, THUS TRYING TO CAPTURE ADHERENCE AND ITS BARRIERS ACROSS ALL LEVELS IN THE MAIN CITY CENTRE OF THE COUNTRY. WE CHOSE TO TARGET THE YOUNGER AGED CHILDREN UNDER THE CARE OF CAREGIVERS RATHER THAN COMBINING THEM WITH ADOLESCENTS AS THEIR BARRIERS DIFFER GREATLY AND THUS OUR RESULTS ARE QUITE INDICATIVE OF THE SITUATION AFFECTING THIS PARTICULAR AGE GROUP.

Key words:

1. Please use key words rather than phrases like ART adherence, associated factors, children, Magnitude…..

RESPONSE: CHANGED TO KEYWORDS RATHER THAN PHRASES

Abstract 

1. Objectives: the objective should be inline with your title i.e. “to assess magnitude and associated factors of ART adherence among children (1-14 years) attending HIV care …

RESPONSE: CHANGED TO THE ABOVE

2. Result: what are all the three measures? Please also approximate the odds ratios and confidence intervals to the nearest number and use only two-digit numbers like 3.520 → 3.5

RESPONSE: THE 3 MEASURES WERE DESCRIBED IN THE METHODS SECTION OF THE ABSTRACT AS FOUR DAY SELF-REPORT, ONE MONTH SELF-RECALL REPORT AND MISSED CLINIC APOINTEMENTS

HAVE APPROXIMATED THEM TO THE NEAREST DECIMAL PLACE

Introduction 

1. First begin by defining HIV/AIDS and adherence

RESPONSE: HAVE INCLUDED THAT

2. Please also state the severity and magnitude of the problem (Poor ART adherence) in Tanzanian context as well. 

RESPONSE: LIMITED STUDIES AVAILABLE FOR THAT ESPECIALLY AMONGST CHILDREN, HAVE INCLUDED WHAT WAS FOUND.

Methods and materials

Study design and area

1. Your title includes 1-14 years but in study design you included 0-14 years. why?

RESPONSE: MY APOLOGIES, THIS WAS A TYPO, HAVE CORRECTED IT.

2. In the paragraph you included children who had regular follow-up. Since you study adherence you have to include children who didn’t have also regular follow up. If you include clients who had regular follow up alone, it over estimates the magnitude of good adherence. 

REPSONSE: THIS WAS A MISINTERPRETATION, PROBABLY A WRONG CHOICE OF WORDS ON MY SIDE; I MEANT CHILDREN WHO ARE ENROLLED IN THAT CLINIC AND SCHEDULED TO ATTEND THERE FOR REGULAR REFILLS AND FOLLOW UP VISITS. NOT THAT I CHOSE PATIENTS WHO REGULARLY ATTENDED. AMONGST THESE PATIENTS WERE THOSE WHO HAD POOR FOLLOW UPS AND THUS DETECTED ON MISSED CLINIC VISIT ADHERENCE MEASURE. HAVE CHANGED IT TO BE MORE CLEAR

Participants and eligibility criteria

1. What is presumptive treatment? Is this recommended by WHO? 

PRESUMPTIVE TREATMENT IS PART OF OUR COUNTRIES NATIONAL HIV/AIDS TREATMENT GUIDELINES AND PART OF OUR PRACTICE WHERE INFANTS AND CHILDREN<18MONTHS BORN TO HIV POSITIVE MOTHERS ARE GIVEN A PRESUMPTIVE DIAGNOSIS OF SEVERE HIV INFECTION BASED ON CERTAIN CRITERIA IN ORDER NOT TO DELAY TREATMENT WHILE AWAITING CONFIRMATORY DNA PCR RESULTS. FURTHER DETAILS CAN BE FOUND FROM THE NATIONAL GUIDELINES FOR THE MANAGEMENT OF HIV/AIDS, 2017, CHAPTER 7, PAGE 111 ( Available in the references).

2. Why you used quota sampling? It is better to use probability sampling techniques in order to generalize your findings. 

RESPONSE: QUOTA SAMPLING WAS USED AS WE WERE TIME BOUND AND HAD A TIGHT BUDGET. 

3. You used clinical hospital records. You have to set some exclusion criteria, if major variables are missed in the chart.

RESPONSE: HOSPITAL RECORDS WERE NOT USED TO COLLECT DATA, AS DATA WAS COLLECTED FROM A STRUCTURED QUESTIONNAIRE DURING AN INTERVIEW PROCESS WITH THE CAREGIVER. THE RECORDS WERE ONLY USED TO CONFIRM THE DIAGNOSIS OF HIV, DRUG REGIMEN THEY ARE ON AND OTHER SUPPORTIVE INFORMATION. THIS ENSURED NO MISSING VARIABLES FOR OUR STUDY.

Variables and data measurement 

1. Better to state your variables in categories of dependent (outcome variable) and independent (predictor) variables in paragraph form

RESPONSE: ADDED THAT AS A PARAGRAPH

2. Adherence measurement: adherence can be adherence to care (clinical adherence) meaning regular attendance of the patient according to given appointment, and drug adherence which means taking the drug according to instructions given by the providers. Your focus is adherence to drug (ART). Therefore, why you used missed clinic appointments as measure of adherence? 

RESPONSE: AGREED, HOWEVER IN OUR SETUP DRUG REFILLS ARE PROVIDED DURING THESE SCHEDULED MONTHLY VISITS, THUS IF A PARTICIPANT DOESNOT ATTEND A CLINIC VISIT THEY WILL MISS A DRUG REFILL AND BE UNABLE TO TAKE THE DRUGS AS SCHEDULED AND INSTRUCTED BY THE PROVIDER AND THUS CONSIDERED AN OBJECTIVE WAY TO ASSESS DRUG ADHERENCE. 

3. It is better to use WHO adherence scale by calculating percent (%) adherence and classify level of adherence as good, fair or poor. You can merge fair and poor based on your operational definition you stated.

RESPONSE: IN THIS STUDY, THE VISUAL ANALOGUE SCALE (VAS) WAS USED TO OBTAIN THE PERCENTAGE ADHERENCE AND CLASSIFIED AS POOR OR GOOD, THIS WAS CHOSEN AS THE CAREGIVER HAD TO OBSERVE THE SCALE AND MARK ON THEIR OWN NOT BEING INFLUENCED OR JUDGED FOR THE MARK THEY PUT THUS REDUCING THE EFFECT OF SOCIAL DESIRABILITY BIAS. FURTHER MORE IT GAVE A DEPICTION OF WHAT THEY PERCEIVED THEIR ADHERENCE WAS, WHICH IS IMPORTANT IN THEIR FUTURE ADHERENCE CHOICES. THIS SCALE HAS BEEN VALIDATED AMONGST STUDIES AND FOUND USEFUL IN RESEARCH AND CLINICAL SETTINGS.

4. In recall report of measuring adherence, how did you handle recall bias?

RESPONSE: IN ORDER TO MINIMISE THE RECALL BIAS WE CAREFULLY SELECTED THE QUESTIONS/ MEANS OF ASKING. USING THE VAS AND RECALLING OVER THE LAST MONTH WHICH WAS NOT TOO LONG AGO, AS IT HAS BEEN SHOWN RECALL BIAS IS PROBLEMATIC WHEN THE EVENT OCCURS LONG TIME AGO. FURTHER MORE WE COMBINED IT WITH SHORTER RECALL METHODS TO REDUCE THE RECALL ELEMENT AND MORE OBJECTIVE MISSED CLINIC VISIT. 

Factors affecting adherence

1. Since it is part of variables, it is better to write in variable section before measures of adherence and outcome.

RESPONSE: I HAVE MOVED IT IN THE VARIABLES SECTION, BUT I FEEL ITS BETTER PUT AFTER MEASURES OF ADHERENCE AS ADHERENCE MEASURING IS THE FIRST OBJECTIVE AND FACTORS AFFECTING ADHERENCE IS THE SECOND OBJECTIVE.

Statistical analysis:

1. What is the importance of chi square test if you use odds ratio? Odds ratio shows measure as well as strength of association. Therefore, it is better to use odds ratio and logistic regression model. 

RESPONSE: APOLOGIES, THAT WAS A TYPO ON MY SIDE, WE USED ODDS RATIO AND LOGISTIC REGRESSION MODELS. THANKS FOR NOTING THAT. I HAVE RECTIFIED THE TYPO.

2. In the logistic regression model, it is better to use bivariable and multivariable logistic regression rather than univariate regression 

RESPONSE: AGREED, BIVARIATE AND MULTIVARIABLE LOGISTIC REGRESSION WAS DONE. 

Result: 

1. What is the importance of mentioning 830, 476 … if this population doesn’t meet inclusion criteria and not part of your study? Focus only on sample size.

RESPONSE: REMOVED THE FLOW DIAGRAM AND EXPLANATION ON RECRUITMENT.

2. Write sociodemographic characteristics before ART adherence among HIV infected children

RESPONSE: DONE

3. What is the importance of writing table 4 and 5, if all of the variables are insignificant?

RESPONSE: MY TAKE ON THIS WOULD DIFFER, THE STUDY OBJECTIVE WAS TO FIND OUT THE FACTORS AFFECTING ART ADHERENCE AMONG CHILDREN, IN TERMS OF THEIR SOCIO-DEMOGRAPHIC CHARACTERISTICS, HEALTH FACILITY/ DRUG RELATED FACTORS. IT WAS ONLY AFTER MULTIVARIABLE LOGISTIC REGRESSION IT LOST SIGNIFICANCE. I THINK ITS IMPORTANT TO SHOW THEY WERE SOUGHT, HOWEVER LOST SIGNIFICANCE AFTER REGRESSION ANALYSIS. 

4. It is better to write the factors affecting adherence in 1 table 

RESPONSE: TABLE 2,3,4,5 IS SHOWING BIVARIATE REGRESSION ANALYSIS OF THESE FACTORS. HOWEVER TABLE 6 IS ONLY DEPICTING THOSE FOUND SIGNIFICANT ON BIVARIATE ANALYSIS, BEING FURTHER ANALYSED IN MULTIVARIABLE LOGISTIC REGRESSION MODEL AND THUS COMING UP WITH THE MOST SIGNIFICANT FACTORS IN ONE TABLE.

Discussion 

1. Paragraph 1: please discuss only your pertinent findings. Paragraph 1 is not related to your study. Avoid it.

RESPONSE: THANKS, REMOVED IT.

2. Please write similar concepts in 1 paragraph in brief and succinct way.

RESPONSE: THANKS HAVE REMOVED THE INSIGNIFICANT FINDINGS AND DWELLED ON WHAT WAS SIGNIFICANT TO THIS STUDY.

3. No need to write sub themes like “Factors affecting ART adherence in children…” in discussion. Please discuss the factors without subheadings. 

REPSONSE : NOTED THANKS

Conclusion:

1. Conclude based on your objective 

RESPONSE : THANKYOU, THE FEEDBACK IS NOTED.

Recommendation:

2. Recommend based on your result 

RESPONSE: THANKYOU, THE FEEDBACK IS NOTED.

Summary status: Major revision

RESPONSE: THANKYOU FOR ALL YR COMMENTS AND CONSTRUCTIVE FEEDBACK. I APRRECIATE THE TIME YOU TOOK TO READ THROUGH AND ADVISE. HAVE REVISED MOST OF THE RECOMMENDED CHANGES, AND GIVEN AN EXPLANATION FOR THE REST.

REVIEWER 2

First, I would like to congratulate the authors for exploring such an important topic. Magnitude and Attributes Of Anti-Retroviral Therapy Adherence Among Children (1-14 Years) Attending HIV Care And Treatment Clinics In Dar Es Salaam, Tanzania. However I have some comments and questions for them, these are found below

Abstract:

Introduction: 

Line 1: In Epidemiological disease level classification HIV is a Pandemic disease; use this term instead of Epidemic. In addition at the end of the introduction please add justification/the need to conduct this study in a line or two. 

RESPONSES: CHANGED THE WORD TO PANDEMIC. ADDED A JUSTIFICATION LINE FOR THE STUDY.

Objectives: Correct the English (Among HIV positive children). Objective must be SMART which includes the “T” which stands for time bound. Therefore specify the time of the study. 

RESPONSES: CORRECTED THE ENGLISH, SPECIFIED THE TIME

Methods: The first statement is not even a full sentence. The method presented is too shallow. The sample size, the method of data analysis, level of significance, the measurement of association, the model fitness assessment, the software used all are lacking. 

RESPONSES: THANKYOU FOR YR CONTRIBUTION; COMPLETED THE SENTENCE. ADDED ALL THE NECESSARY MISSING DETAILS.

Result: The magnitude of good adherence reported (60%) is a point estimate, you should provide the confidence interval. Take the type of analysis used in the method section, use the term multivariable logistic regression instead of multivariate (Multivariate is when you have multiple outcome variables). Age is better reported as a range instead of <25 for children of very young (underage) mothers may not have the same association. Belief in ART effectiveness is not clear (Negative or Positive?? And how did you assess it??). After all the associated factors were written the sentence was concluded as “”… were less likely to be associated poor adherence.” Yet all the significant variables are not negatively associated with the outcome variable. Besides, though, the authors reported the magnitude of good adherence, they reported associated factors for poor adherence, you need to be consistent. 

RESPONSES: WE CANNOT DO PREVALENCE AS WE DONOT HAVE THE ENTIRE DENOMINATOR OR TOTAL NUMBER OF CHILDREN ATTENDING THE FACILITY OVER TIME EG. PER YEAR. WE ONLY COUNTED THOSE WHO CAME DURING THE SAID DATES AND CALCULATED A PROPORTION OUT OF THAT FOR GOOD VERSUS BAD ADHERENCE

Conclusion and recommendations: The authors should re-write this section. Most of it is wrong and the recommendations given are not based on the findings.

RESPONSE: THANKYOU FOR YOUR COMMENT. HAVE CONCLUDED AND RECOMMENDED MY STUDY AS I VISIONED IT TO BE AND TO THE BEST OF MY ABILITY.

Methods and Materials 

Study design and area: In the first line authors mentioned “This was hospital based descriptive cross-sectional study” which is totally wrong. You have assessed factors that affecting adherence therefore it is not mere description. (Use hospital based cross-sectional study) 

RESPONSE: CHANGED AS ADVISED 

Participants and eligibility criteria 

The minimum sample size determination must be clearly presented. There is no such a thing as “quota sampling calculation” Quota sampling is a sampling technique (How you distribute the sample size you determined across quotas). 

How was the three study places identified?? Why none probability sampling???

How can you take consent form under 14 years old children?? (This must be clearly described in the Ethical consideration)

RESPONSES: MY APOLOGIES THAT WAS A TYPO, I MEANT QUOTA SAMPLING WAS DONE. HAVE CLEARLY SHOWN HOW THE SAMPLE SIZE WAS DERIVED AND HOW WERE THEY DISTRIBUTED AMONGST THE 3 SITES.

THE THREE SITES WERE MORE ACCESSIBLE AND REPRESENTED THREE DIFFERENT LEVELS OF HEALTH FACILITIES IN ORDER TO MAKE THE STUDY REPRESENTABLE IN THE CAPACITY WE HAD.

INFORMED CONSENT WAS SOUGHT FROM THE CAREGIVER WHO BROUGHT THE CHILD TO THE CTC CLINIC. A WRITTEN INFORMED ASSENT WAS SOUGHT FROM CHILDREN WHO WERE CAPABLE OF UNDERSTANDING. HAVE CLARIFIED IT IN THE ETHICAL CONSIDERATION SECTION.

Variables and measurements

Clearly put the outcome variable and list of independent variables considered. 

RESPONSES: HAVE CLEARLY PUT IT DOWN.

Statistical analysis 

What method of goodness of fit assessment did you use?? What was its value??

Generally the method part lacks the following: Study period, Sample size determination, Sampling Procedures, Data collection procedure and quality assurance. 

RESPONSE: HAVE ADDED THE DETAILS IN THE METHODOLOGY

Results

The first paragraph must be part of the method section. You should focus on what you have found and not how you did it in the result section. 

The order of the result presentation is full of flaws. First report the characteristics of study participants be it socio-demographic characteristics or co-morbidities. Then go to the descriptive findings of your study which the level/magnitude of adherence, finally you can present the factors associated. The level/magnitude of adherence also needs to be reported using confidence intervals. 

RESPONSE: HAVE REMOVED THE FIRST PARAGRAPH AND ARRANGED THE ORDER AS ADVISED.

Factors associated with ART adherence

Why are the authors reporting the associated factors as socio-demographic and participant related and the like all variables must be adjusted in the multivariable analysis and be reported as one multivariable table. 

RESPONSE: TABLE 2-5 SHOWS THE BIVARIABLE ANALYSIS OF THE FACTORS WE ARE LOOKING INTO CATEGORISED AS SOCIOECONIMIC FACTORS, PARTICIPANT FACTORS, DRUG RELATED FACTORS AND HEALTH SYSTEM RELATED FACTORS. ONLY THOSE WHICH WERE FOUND SIGNIFICANT WERE FURTHER ANALYSED FOR MULTIVARIABLE ANALYSIS AND PUT IN ONE SINGLE TABLE. IT MAY BE DIFFICULT TO PUT ALL THE TABLES IN BIVARIABLE ANALYSIS IN 1 SINGLE TABLE AS WE HAVE CATEGORISED IT. HOWEVER THE MULTIVARIABLE TABLE IS ONE IE: TABLE 6

---

## [Decision Letter · Decision Letter 1]

13 Apr 2021

PONE-D-20-25583R1

MAGNITUDE AND ASSOCIATED FACTORS OF ANTI-RETROVIRAL THERAPY ADHERENCE AMONG CHILDREN ATTENDING HIV CARE AND TREATMENT CLINICS IN DAR ES SALAAM, TANZANIA

PLOS ONE

Dear Dr. Mussa,

Thank you for submitting your manuscript to PLOS ONE. After careful consideration, we feel that it has merit but does not fully meet PLOS ONE’s publication criteria as it currently stands. Therefore, we invite you to submit a revised version of the manuscript that addresses the points raised during the review process.

We look forward to receiving your revised manuscript.

Kind regards,

Satya Surbhi, PhD

Academic Editor

PLOS ONE

Reviewers' comments:

Reviewer's Responses to Questions

**Comments to the Author**

1. If the authors have adequately addressed your comments raised in a previous round of review and you feel that this manuscript is now acceptable for publication, you may indicate that here to bypass the “Comments to the Author” section, enter your conflict of interest statement in the “Confidential to Editor” section, and submit your "Accept" recommendation.

Reviewer #1: All comments have been addressed

Reviewer #2: All comments have been addressed

Reviewer #3: (No Response)

2. Is the manuscript technically sound, and do the data support the conclusions?

Reviewer #1: Yes

Reviewer #2: Yes

Reviewer #3: Partly

3. Has the statistical analysis been performed appropriately and rigorously? 

Reviewer #1: Yes

Reviewer #2: Yes

Reviewer #3: No

4. Have the authors made all data underlying the findings in their manuscript fully available?

Reviewer #1: Yes

Reviewer #2: Yes

Reviewer #3: Yes

5. Is the manuscript presented in an intelligible fashion and written in standard English?

Reviewer #1: Yes

Reviewer #2: Yes

Reviewer #3: (No Response)

6. Review Comments to the Author

Reviewer #1: I would like to appreciate the authors for studying a public health problem. Magnitude and associated factors of Anti-Retroviral Therapy Adherence Among Children (1-14 Years) Attending HIV Care and Treatment Clinics in Dar Salaam, Tanzania)

The response and revision given by the authors for my previous concerns is plausible.

# Discussion

In the discussion section, it better to write an introductory paragraph what your study focuses on (magnitude and factors associated with ART adherence) at the beginning and then proceed to discuss pertinent findings as you did already.

Reviewer #2: The authors have addressed the comments from my reviews in an intelligent and scientific way. I accept that the manuscript is worth publishing.

Reviewer #3: Thank you all the authors for making the requested revisions. I believe you have made a significant improvement from the original manuscript. But I have some issues I asked but not well addressed.

Here are my comments and questions

Abstract

1. The English of the manuscript is still poor. For instance in the method section of the abstract I have commented the first sentence is not even a full sentence yet it is not corrected. It has no verb.

2. In the result I requested the authors to put a confidence interval for the point estimate of adherence level which is 60%. But I don’t think they have got the idea of confidence interval. As long as you determine proportion (which actually is a prevalence) you can always determine the confidence interval for it. You just have to check the boostrap in the SPSS while you run to determine the prevalence (Proportion). If you want to use STATA, you have a total of 200 children with good adherence therefore using the command (cii 333 200) you can find the point estimate to be 60% with 95% CI of (54.58, 65.36).

3. In the previous comment I have also asked the authors to put the age range for caregivers/Mothers instead of putting < 25 for the case may not be the same for very young or teen mothers than the others. It will also has implication in terms of showing the effect of teen pregnancy or motherhood. All the associated factors do not have the same direction but the statement in the result section is closed as ” … less likely to be associated” this is not appropriate even if all are negatively associated the phrase “less likely associated does not give sense”. All these comments were previously given but no improvement is made or no explanation is given.

4. The conclusion given is not based on the results. For example WHO stage of the disease was not significantly associated factor, yet Adherence counseling is recommended for children with advanced disease. You do not just give recommendations by your idea of what should be done, it must be based on the evidence/finding you have.

MAIN PART OF THE MANUSCRIPT

5. In the sample size calculation you should put reference for the study you took the prevalence (24.6%) from. The allocation of study participants in each clinic looks proportional. Therefore it is better put as “The sample was proportionally allocated to the three clinics/hospitals” than quota.

6. In the variables of the study the authors write the Dependent variable to be Adherence whereas the associated factors to be independent variables. How could you just say the associated factors are independent variables?? This is just telling the other name for independent variables. What you are asked is to list the independent variables considered in this study. This comment was previously given as well.

7. In the data processing and analysis section the question of goodness of fit was raised but it was not addressed. P-value tells us if the variable is significantly associated with the outcome, it does not tell us if the model used appropriate for the data. We use the Hosmer-Lemshow goodness of fit test to show this in logistic regression. This was not addressed.

8. In the result section: if the sample size is 316 why 333 people are included in the study? The table showing the multivariable outcome (Table 6) should also include the crude odds ratios.

7. PLOS authors have the option to publish the peer review history of their article (what does this mean?). If published, this will include your full peer review and any attached files.

Reviewer #1: No

Reviewer #2: No

Reviewer #3: **Yes: **Adhanom Gebreegziabher Baraki

---

## [Author Response · Author response to Decision Letter 1]

8 Jul 2021

THANKYOU ALL FOR YR VALID COMMENTS.

REVIEWER 1:

1. THANKS FOR YOUR COMMENT, IT WAS PREVISOULSY COMMENTED I SHOULD REMOVE THE INTRODUCTORY COMMENT AND JUMP STRAIGHT TO DISCUSSING MY RESULTS AND THUS I HAD DELETED IT. HOWEVER, I HAVE AGAIN INCLUDED A SHORT INTROCUTORY COMMENT AS SUGGESTED IN THE DISCUSSION SECTION.

REVEIWER 3: 

THANKYOU FOR YR COMMENTS AND DETAILED CLARIFICATION, I HAVE TRIED TO CLARIFY AND ANSWER TO MY BEST CAPABILITIES.

1. IT WAS AN OVERSIGHT ON MY PART THAT THE VERB WAS MISSING IN THE METHOD SECTION OF THE MANUSCRIPT, I HAVE NOW INCLUDED IT AND HAVE TRIED MY BEST TO PERFECT THE ENGLISH IN THE BEST OF MY ABILITIES.

2. CONFIDENCE INTERVAL FOR THE GOOD ADHERENCE: I HAVE ADDED IT, CALCULATED IT FROM SPSS AND PUT IT IN THE RESULT SECTION. 

3. AGE RANGE<25 CAREGIVERS, WERE 17-25 YEARS, OF WHICH ONLY 7 PARTICIPANTS (2.7%) WERE 17-19 YEARS, THE REST ( 15, ie: 5.4%) WERE 20-24 YEARS.

4. LOWER CLINICAL STAGE (1-2) AS COMPARED TO ADVANCED STAGES (3-4) WERE SEEN TO BE LESS LIKELY ASSOCIATED WITH POOR ADHERENCE. IF YOU RUN THE STATISTICS VICERVERSA IT WILL SHOW THAT ADVANCED STAGES WILL MORE LIKELY BE ASSOCOATED WITH POOR ADHERENCE. THUS, IT SEEMS FAIR TO CONCLUDE THAT CHILDREN WITH ADVANCED STAGES NEED MORE ADHERENCE COUNSELLING TO MY UNDERSTANDING. I WAS TRYING TO KEEP THE LANGUAGE STANDARD WHERE WE COMPARE THE FACTORS AGAINST POOR ADHERENCE RATHER THAN SOME FACTORS AGAINST POOR ADHERENCE AND SOME AGAINST GOOD ADHERENCE.IT WOULD SOUND INAPPROPRIATE TO SAY LOWER CLINICAL STAGE WAS MORE LIKELY ASSOCOATED WITH GOOD ADHERENCE AS THE PREVIOUS FACTORS WERE DESCRIBED IN RELATION TO POOR ADHERENCE.

5. REFERENCE FOR STUDY FOR SAMPLE SIZE- THAT HAS BEEN STATED IN THE MAIN PART OF THE MANUSCRIPT WITH THE REFERENCE GIVEN. IN REGARDS TO THE SAMPLE SIZE ALLOCATION- ALTHOUGH IT SEEMS LIKE THE ALLOCATION OF PARTICIPANTS WERE PROPORTIONAL, IT WAS STILL QUOTA SAMPLING THAT WAS PERFORMED TO GET EXACT NUMBERS AS EXPLAINED. IT WILL RAISE A QUESTION OF WHY THE SAMPLE WAS NOT EXACTLY IDENTICAL IF WE MENTION THE SAMPLE WAS PROPORTIONALLY ALLOCATED. DUE TO THE ALMOST NEAR NUMBER OF PARTICIPANTS REGISTERED FROM EACH CLINIC SITE, IT TURNED OUT TO BE ALMOST PROPORTIONAL. 

6. INDEPENDENT VARIABLES: THE INDEPENDENT VARIABLE BY DEFINITION IS THE VARIABLE THE EXPERIMENTER MANIPULATES OR CHANGES AND IS ASSUMED TO HAVE A DIRECT EFFECT ON THE DEPENDANT VARIABLE. IN THIS CASE OUT DEPENDANT VARIABLE OR OUTCOME IS THE ADHERENCE. SO, WHAT INFLUENCES THE ADHERENCE IN THIS CASE ARE THE FACTORS (DERIVED FROM LITERATURE REVIEW) THAT MAY OR MAY NOT CHANGE THE OUTCOME. YES, NOT ALL ASSOCIATED FACTORS ARE INDEPENDENT VARIABLES BUT, IN THIS STUDY ALL THESE FACTORS ARE CAPABLE OF AFFECTING THE OUTCOME VARIABLE. THE AUTHORS REQUEST THE REVIEWER TO GIVE US A FURTHER EXPLANATION CONTRARY TO THIS AND EXPLAIN WHAT HE UNDERSTANDS BY THE TERM INDEPENDENT VARIABLES. THANKYOU.

7. GOODNESS OF FIT

8. SAMPLE SIZE WAS 316, WHICH IS THE MINIMAL SAMPLE SIZE CALCULATED, HOWEVER WE MANAGED TO COLLECT DATA FOR MORE CHILDREN (313) WITHIN THE TIME FRAME AND COULD NOT REFUSE THE PATIENTS TO ENROLL IN THE STUDY. THUS, INCLUDED THEM THE ANALYSIS. IN ADDITION, WE HAVE NOT EXCEEDED THE 10% OF THE ALLOWANCE FOR THE SAMPLE SIZE. HAVING LESS PARTICIPANTS WOULD WEAKEN THE POWER OF THE STUDY, REACHING AND EXCEEDING SAMPLE SIZE INCREASES THE POWER OF THE STUDY. 

9. CRUDE ODDS RATIO: INCLUDED AS REQUESTED

---

## [Decision Letter · Decision Letter 2]

1 Dec 2021

PONE-D-20-25583R2MAGNITUDE AND ASSOCIATED FACTORS OF ANTI-RETROVIRALTHERAPY ADHERENCE AMONG CHILDREN ATTENDING HIV CARE AND TREATMENT CLINICS IN DAR ES SALAAM, TANZANIAPLOS ONE

Dear Dr. Mussa,

Thank you for submitting your manuscript to PLOS ONE. After careful consideration, we feel that it has merit but does not fully meet PLOS ONE’s publication criteria as it currently stands. Therefore, we invite you to submit a revised version of the manuscript that addresses the points raised during the review process.

We look forward to receiving your revised manuscript.

Kind regards,

Satya Surbhi, PhD

Academic Editor

PLOS ONE

Journal Requirements:

Reviewers' comments:

Reviewer's Responses to Questions

**Comments to the Author**

1. If the authors have adequately addressed your comments raised in a previous round of review and you feel that this manuscript is now acceptable for publication, you may indicate that here to bypass the “Comments to the Author” section, enter your conflict of interest statement in the “Confidential to Editor” section, and submit your "Accept" recommendation.

Reviewer #2: All comments have been addressed

Reviewer #3: (No Response)

2. Is the manuscript technically sound, and do the data support the conclusions?

Reviewer #2: Yes

Reviewer #3: Partly

3. Has the statistical analysis been performed appropriately and rigorously? 

Reviewer #2: Yes

Reviewer #3: Yes

4. Have the authors made all data underlying the findings in their manuscript fully available?

Reviewer #2: Yes

Reviewer #3: No

5. Is the manuscript presented in an intelligible fashion and written in standard English?

Reviewer #2: Yes

Reviewer #3: No

6. Review Comments to the Author

Reviewer #2: (No Response)

Reviewer #3: Thank you, all the authors, for making the requested revisions. I believe you have made improvement from R1. But I have some issues I asked but not well addressed. Here are my comments and questions.

General Comment:

I think it could be better if the authors give the manuscript to a native English speaker or language expert to do proof reading. The overall quality of the language used is still poor.

2. You added the confidence interval in the result section of the main manuscript only put it in the abstract too.

3. The age range 17-25 years was mentioned in the table but not in the other parts of the manuscript like the abstract. When you do corrections in one part try to make it consistent across all parts of the manuscript.

4. In the result section of the abstract. All the associated factors do not have the same direction but the statement in the result section is closed as ” … less likely to be associated” this is not appropriate even if all are negatively associated. The phrase “less likely associated” is not the right way to interpret odds ratio. I recommend reading more on interpretation of Odds ratios.

Main manuscript:

METHOD

5. I am afraid the authors get the philosophy behind quota sampling. The participants are taken from each institution either proportionally or in equal number in this study but not using quota sampling. Quota sampling is a non-probability sampling, and the participants are selected based on specific qualities and traits. There is no such considerations here.

6. The definition of what dependent variable is and what independent variable is not needed here. You just have to mention what the dependent variable is and what independent variables are considered in this study. For instance you can say: The dependent variable in this study was Adherence where as the independent variables were socio-demographic characteristics like age, gender, marital status . . .; Treatment related factors like type of regimen, duration on ART. . .

7. The authors just write the phrase “Goodness of fit” in their response about how the goodness of fit of the model was addressed. In the result section it says 0.243 for the P-value of the model fitness test. I recommend to mention Hosmer-Lemshow goodness of fit was used to test the association.

8. Including sample of study participants above the minimum may not have weaken the study. Actually, it increases the power. The thing is participants are not supposed to be included just because they wanted to participate. This is why we needed the sample size and sampling procedure. 10% is also to replace non-responders and it is considered in the final sample size; It is not in addition to the final sample size you have. The authors need to give scientific justification as this may affect rigor in conducting research and its replicability.

9. The final multi-variable table shall have all the sections provided below

Variable | Adherence (frequency and percentage for both good and poor) | COR(95% CI) | AOR (95% CI)

There is also no need to write P-values for COR and AOR. P-value for AOR is enough or you can use * for P-value less than 0.05 and mention this as a footnote.

7. PLOS authors have the option to publish the peer review history of their article (what does this mean?). If published, this will include your full peer review and any attached files.

Reviewer #2: No

Reviewer #3: **Yes: **Adhanom Gebreegziabher Baraki

---

## [Author Response · Author response to Decision Letter 2]

26 Jan 2022

I am thankful to the entire team for their comments and contributions to this manuscript.

These are the responses to the questions posed by Adhanom Gebreegziabher Baraki.

1) I have sent the work to a native English speaker to proofread. I hope it is satisfactory as it stands.

2/3) I have added the confidence interval for adherence and changed the age range 17-25 in the abstract and throughout the manuscript.

4) Pertaining to the direction to interpretation of the negatively associated odds ratio, we have changed the reference variable and made it a positive odds ratio with a more harmonized interpretation. Thus made a change to those 2 variables ie: belief in ART effectiveness and clinical stage variables. 

From my reading, odds ratio may be interpreted as odds of having an event occurring was…….. times more or less likely in the exposed group than in the unexposed.

5) Regarding the sampling procedure I have understood what you meant, I have rectified it to Proportion to size sampling rather than quota sampling and explained the technique used in the manuscript.

6) I have indicated the dependent and independent variables considered in this study as requested.

7) In regards to Goodness of fit, I had specified in the results section the Hosmer-Lemshow goodness of fit was used. However I have rephrased it again as requested.

8) Regarding the sample size, I understand that participants are not supposed to be included because they wanted to participate. I apologize for the over enrollment that took place as we were two interviewers collecting simultaneously at different sites, So that led to over recruitment due to lack of communication. Once we collected the data, we felt it unwise to discard the data. Thankyou for pointing that out.

9) The final multivariate table has been re-visited and all the variables required in cooperated.

---

## [Decision Letter · Decision Letter 3]

16 Aug 2022

PONE-D-20-25583R3MAGNITUDE AND ASSOCIATED FACTORS OF ANTI-RETROVIRALTHERAPY ADHERENCE AMONG CHILDREN ATTENDING HIV CARE AND TREATMENT CLINICS IN DAR ES SALAAM, TANZANIAPLOS ONE

Dear Dr. Mussa,

Thank you for submitting your manuscript to PLOS ONE. After careful consideration, we feel that it has merit but does not fully meet PLOS ONE’s publication criteria as it currently stands. Therefore, we invite you to submit a revised version of the manuscript that addresses the points raised during the review process. Please note some reviewers may have made edits/comments to the manuscript in addition to their comments and responses on the review form. Kindly review and respond to all their comments.

We look forward to receiving your revised manuscript.

Kind regards,

Chika Kingsley Onwuamah, Ph.D.

Academic Editor

PLOS ONE

Journal Requirements:

Reviewers' comments:

Reviewer's Responses to Questions

**Comments to the Author**

1. If the authors have adequately addressed your comments raised in a previous round of review and you feel that this manuscript is now acceptable for publication, you may indicate that here to bypass the “Comments to the Author” section, enter your conflict of interest statement in the “Confidential to Editor” section, and submit your "Accept" recommendation.

Reviewer #2: (No Response)

Reviewer #4: (No Response)

Reviewer #5: (No Response)

2. Is the manuscript technically sound, and do the data support the conclusions?

Reviewer #2: Partly

Reviewer #4: Yes

Reviewer #5: Yes

3. Has the statistical analysis been performed appropriately and rigorously? 

Reviewer #2: Yes

Reviewer #4: Yes

Reviewer #5: Yes

4. Have the authors made all data underlying the findings in their manuscript fully available?

Reviewer #2: Yes

Reviewer #4: No

Reviewer #5: Yes

5. Is the manuscript presented in an intelligible fashion and written in standard English?

Reviewer #2: Yes

Reviewer #4: Yes

Reviewer #5: Yes

6. Review Comments to the Author

Reviewer #2: The author has addressed the comments previously provided by the reviewers. However, there are still issues that the author must address in order to get a sound research work worth publishing. My additional comments are the below:

• Better to put AOR which are significant (i.e., p<=0.05) instead of OR as final results in the results section for factors.

• The AOR can be presented as (AOR=xx, 95% CI[xx,yy]) or (AOR=xx, 95%CI(xx-yy)) after the variable under discussion. The p-value is not needed here as the confidence interval speaks if the test is significant or not.

• In the conclusion section, factors contributing to low adherence from Table 6 must be mentioned (like age of care giver, child inter-current illness, believes on effectiveness of ART, and clinical stage).

• Usually, sample sizes are denoted by n and population by N. So, better to denote your sample size by n.

Reviewer #4: Methodology Concerns:

1. Study Design and study area:

Reference was made to National guidelines. It is useful to state the country of origin of the guidelines. Is this the Tanzanian National guidelines? Please clarify.

2. Methodology is silent on the specific factors affecting adherence only to surface in the results. Unless this is the journal preference, mention should be made in methodology of the specific factors affecting ART adherence.

3. The description of the quota sampling technique was inadequate. This requires more detail to understand the populations of the patients at the CTC sites and how representative the sampled research participants are of the CTC sites. Given the drawbacks of Quota Sampling methodology – Non-probability sampling, researcher’s bias in selection of study participants and limitations with generalization of study findings, a lot more detail would be required. Was the sampling controlled or not controlled for instance.

Results:

4. Socio-demographic factors affecting adherence:

a. Child age grouping of 1-9 years and 10-14 years lumps children with different needs together. The Under—five children age group is entirely dependent on the care giver for medication administration as opposed to the older age groups. School age children attending day school would have different challenges when compared with children in boarding school. Combining all these children together into just two groups with different needs may hide information that is inimical to adherence. Consider age groups of U5, 6-10, 11-14.

b. Similarly, parental grouping could be further revised to Single parents, both parents, grandparents and other caregivers. The challenges for each group differ from the other and may impact differently on adherence.

c. Care giver marital status also appears to be lumped together. Married, cohabiting, single, Divorced, widowed caregivers could be assessed separately. Married and cohabiting on one hand and single divorced and widowed caregivers on the other hand may not have the same challenges. It is possible that evaluating each group may be more informative that combining them.

Conclusion:

5. Pill burden

Specify the type of pill burden, high or low.

Recommendation:

6. Adherence counseling:

The authors’ results show that there is no association between lack of adherence counseling and poor ART adherence and yet recommended the necessity of counseling. This needs clarification.

Reviewer #5: This is a good and generally well-written manuscript.

However, I noted some points that need to be addressed.

1. Line 4 of the 2nd paragraph of the introduction says 1.7 million children (1-14 years) were living with HIV and 1.9 million of them were in sub-Saharan Africa. I think this is a typo as the number is about 1.09 million (64% of the 1.7 million).

2. In statistical analysis, it is mentioned that measures of central tendency were used to summarize discrete data; the only measure of central tendency that can be applied to discrete data is mode and this should be stated if it was used.

3. In the results section: for age groups, 25 years appears in 2 strata: 17-25 years and 25-34 years. This should be appropriately revised; either 17 -<25 years and 25-35 years, OR 17-25 years and >25 - 34 years.

7. PLOS authors have the option to publish the peer review history of their article (what does this mean?). If published, this will include your full peer review and any attached files.

Reviewer #2: No

Reviewer #4: No

Reviewer #5: **Yes: **Agatha N. David

---

## [Author Response · Author response to Decision Letter 3]

5 Sep 2022

Dear editor and reviewer, Thankyou for taking up this review process again. As you may have already known, this work had gone through an extensive review process over the past one and half year, I’m not sure which version of the work you have finally reviewed as some of the comments had already been addressed and edited in the last version. However, I have tried to answer your valid comments in the best of my capacity. Im proud to claim that my work has been validated by 6 reveiwers and 2 editors!!!

Reviewer #2’comments

Comment 1: Better to put AOR which are significant (i.e., p<=0.05) instead of OR as final results in the results section for factors- 

Response : have addressed the suggested valid comments in the manuscript.

Comment 2: In the conclusion section, factors contributing to low adherence from Table 6 must be mentioned (like age of care giver, child inter-current illness, believes on effectiveness of ART, and clinical stage). 

Response: it was already mentioned in the conclusion of the discussion section but was missing in the abstract, I have now included it. 

Comment 3: Usually, sample sizes are denoted by n and population by N. So, better to denote your sample size by n.

Resposnse: Have denoted the sample sizes as n

Reviewer #4 comments

comment 1: Study Design and study area:

Reference was made to National guidelines. It is useful to state the country of origin of the guidelines. Is this the Tanzanian National guidelines? Please clarify.

Response: Yes, I meant the Tanzanian National guidelines for HIV/AIDS. I have clarified it in the document.

comment 2: Methodology is silent on the specific factors affecting adherence only to surface in the results. Unless this is the journal preference, mention should be made in methodology of the specific factors affecting ART adherence.

Response: This comment was already addressed in the revised latest version which I assume you may not have seen, I have highlighted in the document again.

Comment 3: The description of the quota sampling technique was inadequate. This requires more detail to understand the populations of the patients at the CTC sites and how representative the sampled research participants are of the CTC sites. Given the drawbacks of Quota Sampling methodology – Non-probability sampling, researcher’s bias in selection of study participants and limitations with generalization of study findings, a lot more detail would be required. Was the sampling controlled or not controlled for instance.

Response: this also was addressed in the previous review and after discussion, it was concluded that it was not a quota sampling (a misnomer) but rather a proportion to size sampling technique based on the number of patients enrolled in the specific site in order to obtain a representative yet unbiased sample based on the site of recruitment, where participants where then consecutively recruited till the number was reached from each site and eventually the required sample size. I added a better clarification in the manuscript.

Comment 4: Results

Socio-demographic factors affecting adherence: a. Child age grouping of 1-9 years and 10-14 years lumps children with different needs together. The Under-five children age group is entirely dependent on the caregiver for medication administration as opposed to the older age groups. School-age children attending day school would have different challenges when compared with children in boarding school. Combining all these children together into just two groups with different needs may hide information that is inimical to adherence. Consider age groups of U5, 6-10, 11-14.

b. Similarly, parental grouping could be further revised to Single parents, both parents, grandparents and other caregivers. The challenges for each group differ from the other and may impact differently on adherence.

c. Care giver marital status also appears to be lumped together. Married, cohabiting, single, Divorced, widowed caregivers could be assessed separately. Married and cohabiting on one hand and single divorced and widowed caregivers on the other hand may not have the same challenges. It is possible that evaluating each group may be more informative that combining them.

Response: Thank you for this comment, in regards to the child age group we had considered this among many other sub-categorization methods, however during our data collection process we found that under 9 years in our setting were totally dependent on the caregiver for their overall needs and HIV care specifically such that even drugs were being administered solely by the caregiver, even boarding school options were not there for this category. Thus, we deemed it wise to categorize them as we have done due to the similarities. Generally, in this setting children below 10 years are still considered to be totally dependent on their caregivers.

In regards to parental status, we thought it would be wise to look into whether a parental presence would affect the outcome of interest, whether one or both or none as we know a parent would strive hard for their own child as compared to others even if it is a single parent. In our setting, it is quite common to have different partners, married or not at different times, so having another partner instead of the child’s own another parent would not change the outcome and it might be culturally not acceptable to ask if you are still living with the father/mother of the index child, especially if they had no short-term or long-term relationship which is very customary here. Similarly, we opted to group the marital status in a culturally unoffending way as married/cohabiting as we know atleast they have a partner support versus the single/divorced/widowed with no partner support. Cohabiting is culturally accepted here and it may be hostile to ask them whether they’re living together out of wedlock or they have actually married each other. 

Conclusion:

Comment 5. Pill burden

Specify the type of pill burden, high or low.

Response: have specified it as high, thankyou.

Recommendation:

comment 6. Adherence counseling: The authors’ results show that there is no association between lack of adherence counseling and poor ART adherence and yet recommended the necessity of counseling. This needs clarification.

Response: ART adherence was still found to be sub-optimal at around 60%, the most important strategy to improve ART adherence is to strengthen counselling, which in our study is reported to be done, however, may still need to be reinforced in the high risk groups like younger aged caregivers, sick children as these children are still dependant of their caregivers. Younger aged caregivers being less mature than their counterparts may need more support and education. Sick children and those with advanced HIV stages may need to be admitted and have closer follow-ups with their clinicians to improve their condition and improve adherence. Health talks during clinic visits and on social media platforms may improve belief and alleviate misconceptions in the ARTs. There’s no single way to predict patients’ adherence or nonadherence status and it has been seen that adherence is always better earlier in the disease in patients who are more motivate and less fatigued, it is ongoing adherence measurements and early identification of faltering adherence and continuing counselling that may be able to bring it back up. Thus these may not be separable and is vital for ART adherence success. 

Reviewer #5: 

comment 1: Line 4 of the 2nd paragraph of the introduction says 1.7 million children (1-14 years) were living with HIV and 1.9 million of them were in sub-Saharan Africa. I think this is a typo as the number is about 1.09 million (64% of the 1.7 million).

Response: Amends has been made, thanks.

comment 2. In statistical analysis, it is mentioned that measures of central tendency were used to summarize discrete data; the only measure of central tendency that can be applied to discrete data is mode and this should be stated if it was used.

Response: Age in this study was taken in years not considering the months, and median was applied to it to find the median age of children as our age outliers. Mode was used by default for all categorical data.

comment 3. In the results section: for age groups, 25 years appears in 2 strata: 17-25 years and 25-34 years. This should be appropriately revised; either 17 -<25 years and 25-35 years, OR 17-25 years and >25 - 34 years.

This has been addressed and changed accordingly, thank you.

---

## [Editor Report · Decision Letter 4]

19 Sep 2022

MAGNITUDE AND ASSOCIATED FACTORS OF ANTI-RETROVIRALTHERAPY ADHERENCE AMONG CHILDREN ATTENDING HIV CARE AND TREATMENT CLINICS IN DAR ES SALAAM, TANZANIA

PONE-D-20-25583R4

Dear Dr. Mussa,

We’re pleased to inform you that your manuscript has been judged scientifically suitable for publication and will be formally accepted for publication once it meets all outstanding technical requirements.

Kind regards,

Chika Kingsley Onwuamah, Ph.D.

Academic Editor

PLOS ONE
---

## [Editor Report · Acceptance letter]

21 Sep 2022

PONE-D-20-25583R4 

Magnitude and associated factors of anti-retroviral therapy adherence among children attending HIV care and treatment clinics in Dar es Salaam, Tanzania 

Dear Dr. Mussa:

I'm pleased to inform you that your manuscript has been deemed suitable for publication in PLOS ONE. Congratulations! Your manuscript is now with our production department. 

Kind regards, 

on behalf of

Dr. Chika Kingsley Onwuamah 

Academic Editor

PLOS ONE